# *ThetaEvolve*: Test-time Learning on Open Problems

Yiping Wang [1 2]   Shao-Rong Su [1]   Zhiyuan Zeng [1]   Eva Xu [1]   Liliang Ren [2]   Xinyu Yang [3 2]   Zeyi Huang [4 2]
Xuehai He [2]   Luyao Ma [5]   Baolin Peng [2]   Hao Cheng [2]   Pengcheng He [2]   Weizhu Chen [2]   Shuohang Wang [2]
Simon Shaolei Du[† 1]   Yelong Shen[† 2]

## Abstract

Recent advances in large language models (LLMs) have enabled breakthroughs in mathematical discovery, exemplified by AlphaEvolve, a closed-source system that evolves programs to improve bounds on open problems. However, it relies on ensembles of frontier LLMs to achieve new bounds and is a pure inference system that models cannot internalize the evolving strategies. We introduce *ThetaEvolve*, an open-source framework that simplifies and extends AlphaEvolve to efficiently scale both in-context learning and Reinforcement Learning (RL) at test time, allowing models to continually learn from their experiences in improving open optimization problems. ThetaEvolve features a single LLM, a large program database for enhanced exploration, batch sampling for higher throughput, lazy penalties to discourage stagnant outputs, and optional reward shaping for stable training signals, etc. ThetaEvolve is the first evolving framework that enable a small open-source model, like DeepSeek-R1-0528-Qwen3-8B, to achieve new best-known bounds on open problems (circle packing and first auto-correlation inequality) mentioned in AlphaEvolve. Besides, across two models and four open tasks, we find that ThetaEvolve with RL at test-time consistently outperforms inference-only baselines, and the model indeed learns evolving capabilities, as the RL-trained checkpoints demonstrate faster progress and better final performance on both trained target task and other unseen tasks. We release our code publicly.[1]

---

This work was done during Yiping's internship. [1]University of Washington [2]Microsoft [3]Carnegie Mellon University [4]University of Wisconsin-Madison [5]University of California, San Diego. Correspondence to: Yiping Wang <ypwang61@cs.washington.edu>, Simon Shaolei Du <ssdu@cs.washington.edu>, Yelong Shen <yeshe@microsoft.com>.

*Proceedings of the $43^{rd}$ International Conference on Machine Learning*, Seoul, South Korea. PMLR 306, 2026. Copyright 2026 by the author(s).

[1]https://github.com/ypwang61/ThetaEvolve

## 1. Introduction

| | Model | CP(↑) | FACI(↓) |
|---|---|---|---|
| Human | - | 2.634 | 1.5098 |
| AlphaEvolve | Gemini-2.0-Flash/Pro | 2.63586276 | 1.503164 |
| ShinkaEvolve | Claude-sonnet-4/o4-mini/... | 2.63598283 | - |
| **ThetaEvolve** | Distill-Qwen3-8B | 2.63598**308** | 1.503**133** |

*Table 1.* **Improving bounds achieved with ThetaEvolve based on DeepSeek-R1-0528-Qwen3-8B** (DeepSeek-AI, 2025). We consider two open tasks, circle packing (CP) and the first autocorrelation inequality (FACI), and report the best values mentioned in AlphaEvolve-v2 (Georgiev et al., 2025) and its variant ShinkaEvolve (Lange et al., 2025). Notably, the circle-packing program discovered by ThetaEvolve takes **only 3 seconds** to consistently find the same best solution, which is significantly faster than the program found by ShinkaEvolve (around 75 seconds). See Appendix F.2 for details. We also obtain results close to AlphaEvolve on several other tasks (Sec. 4.2).

The recent development of the reasoning capabilities of large language models (LLMs) has enabled them to contribute to new scientific findings, like mathematical discovery (Romera-Paredes et al., 2024; Charton et al., 2024; Wagner, 2021; Fawzi et al., 2022). A notable recent example is AlphaEvolve (Novikov et al., 2025; Georgiev et al., 2025), which uses pre-designed evaluators together with frontier LLMs to iteratively modify and improve candidate programs toward optimizing task-specific objectives. Through this evolutionary process, AlphaEvolve has discovered solutions that match or improve the best-known results for several open mathematical optimization problems. AlphaEvolve is well-suited for problems that aim to construct specific mathematical objects to improve their certain quantitative properties, such as arranging a fixed number of circles within a unit square to maximize the sum of radii (Friedman, 2012) (referred to as `CirclePacking` in our work).

In detail, AlphaEvolve maintains a program database that stores high-scoring or diversity-promoting programs (e.g., those using different strategies) discovered throughout the evolutionary trajectory. At each iteration, AlphaEvolve samples several prior programs from this database to construct a prompt, which is then fed to an ensemble of LLMs to generate improved child programs. These child programs are subsequently evaluated and added back into the program

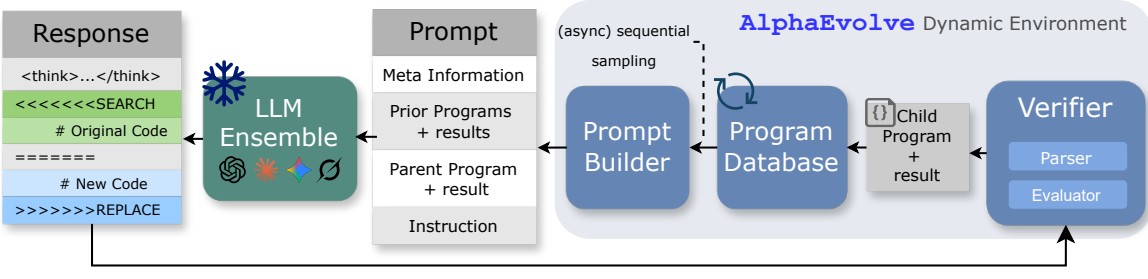

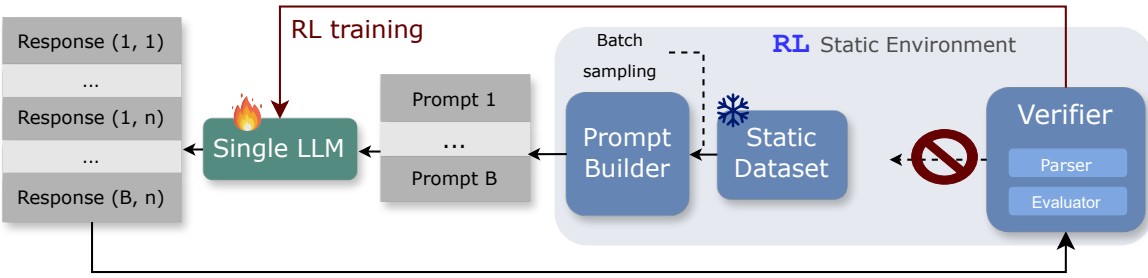

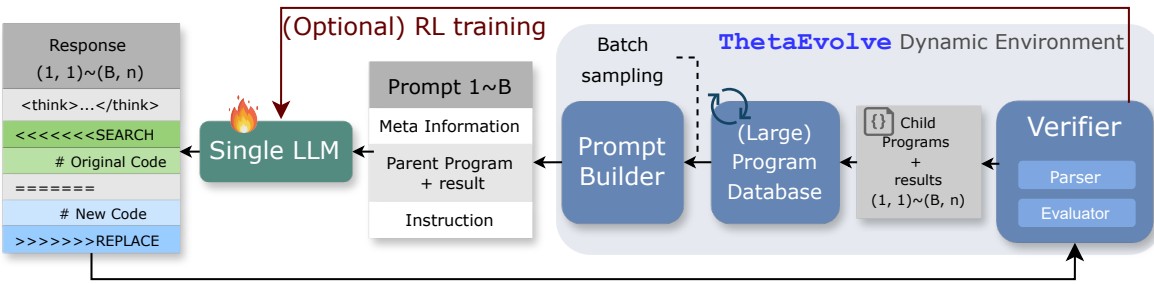

*Figure 1.* **ThetaEvolve draws insights from both the AlphaEvolve pipeline and conventional RL pipelines.** (**Top**) AlphaEvolve/OpenEvolve Dynamic Environment (inference only). (**Middle**) RL Static Environment. (**Bottom**) ThetaEvolve Dynamic Environment (with or without RL training). ThetaEvolve simplifies AlphaEvolve by using a single LLM and (optionally) including only the parent program in the prompt. It adopts a large program database and uses batch sampling at each step to better scale test-time compute. It also incorporates lazy penalties and reward shaping for (optional) RL training.

database (see Fig. 1, top). As we scale the test-time compute, AlphaEvolve can continually learn from its own frontier attempts on open problems, while avoiding unbounded growth in context length.

Nevertheless, AlphaEvolve and its follow-up work also exhibit clear limitations. First, AlphaEvolve remains a closed-source system, which makes systematic study of program evolution on open problems relatively under-explored. Although recent efforts have produced open-source variants such as OpenEvolve (Sharma, 2025) and ShinkaEvolve (Lange et al., 2025), these pipelines are still complex, with many hyperparameters that are not fully ablated, leaving it unclear which components are truly essential. Second, in existing empirical implementations, these pipelines are almost always paired with frontier, large-scale, closed-source LLM ensembles. This implies a mindset that smaller open-

source models, which are more suitable for open research and local deployment, cannot help push the best-known results on these challenging tasks. More importantly, AlphaEvolve is purely an inference-time pipeline and does not update the underlying model at all. Its performance relies entirely on the design of the inference procedure, meaning that effective exploration strategies or "search-on-the-edge" behaviors cannot be learned by the model itself.

On the other hand, reinforcement learning (RL) has demonstrated strong potential for improving reasoning language models (Gao et al., 2024; Lambert et al., 2024; OpenAI, 2024; DeepSeek-AI, 2025; Team et al., 2025; Google DeepMind, 2025). AlphaProof (Hubert et al., 2025) further shows that when the target task is equipped with a self-contained, rule-based verifier such as LEAN, scaling test-time RL can boost performance beyond standard inference-time scaling.

Notably, program-evolution pipeline such as AlphaEvolve share the same structure: once a candidate program is produced, the fixed evaluator can deterministically check validity and compute an objective value for further optimization. Building on this observation, we integrate the evolution on open optimization problems with an RL training pipeline, leading to our framework *ThetaEvolve*. We summarize our contributions below:

(1) We propose a new open-source pipeline, for scaling test-time compute using either pure inference or reinforcement learning (RL) on challenging open problems. To achieve more efficient and effective inference, we introduce several modifications, such as simplifying the LLM ensemble to a single LLM, sampling a batch of parent programs and responses at each step to improve inference throughput (Sec. 4.4.2), and significantly scaling the size of the program database to obtain better final performance (Sec. 4.4.1), etc. To further enable effective test-time RL on open problems, we incorporate a lazy penalty to discourage repeatedly outputting previously strong programs without attempting improvement, and add optional reward shaping to keep training rewards within a reasonable range (Sec. 3, 4.4).

(2) Surprisingly, we show that when scaling test-time compute with ThetaEvolve, a single open-source 8B model, DeepSeek-R1-0528-Qwen3-8B (DeepSeek-AI, 2025), can improve the best-known bounds of two open problems considered in AlphaEvolve: circle packing and the first autocorrelation inequality (Tab. 1), whereas the previous results in AlphaEvolve were achieved using ensembles of strong LLMs such as Gemini-2.0-Flash/Pro. Notably, the circle-packing program discovered by ThetaEvolve takes only 3 seconds to find the current best solution, which is substantially faster than the program found by ShinkaEvolve (around 75 seconds), which uses an ensemble of six advanced closed-source models including Claude-Sonnet-4 and o4-mini (Sec. 4.2)

(3) We further find that using RL with ThetaEvolve consistently outperforms inference-only runs across two open-source models and four challenging problems We verify that the model indeed internalizes nontrivial capabilities for improving evolution performance: when using a checkpoint trained with RL under ThetaEvolve for pure inference on the same task, it achieves better scores and significantly faster progress compared with the original model. This improvement even transfers to other problems, indicating that RL with ThetaEvolve can generalize evolutionary capability across tasks. We also shows that such improvements cannot be obtained by using only a format reward or by performing RL in a static environment (Sec. 4.3).

## 2. Preliminary: AlphaEvolve Pipeline

In this section, we briefly introduce the framework of AlphaEvolve (Novikov et al., 2025; Georgiev et al., 2025) and its open-source implementation, OpenEvolve (Sharma, 2025) (Fig. 1 Top). More details are in Appendix D.

**Manual Preparation.** First, for the target task we aim to optimize, AlphaEvolve requires designing an unhackable evaluator that maps solutions to scalar scores. These systems also require an initial program which provides an example that specifies the basic evaluation format. Moreover, We need meta-information that describes the problem and outlines possible directions for improving existing bounds. AlphaEvolve-v2 demonstrates that the advice provided in the prompt can significantly influence the final performance (Georgiev et al., 2025). The prompts used in the paper are detailed in Appendix C.3.

**Program Database.** During the evolutionary procedure, AlphaEvolve continually generates new programs with their evaluation results attached. They are added into an evolutionary program database, whose purpose is to resample previously explored high-quality candidates for future generations. AlphaEvolve mentions a relatively complex evolutionary algorithm to manage the programs store in the database (See Appendix D.4 for details). In our paper, we focus on ablate the parameters related to database size, like `population_size`, which denotes the maximum number of programs that can be stored. When new programs are added into the program database, database would rank the program based on metrics like objective score or diversity, and delete some programs if the database is full.

**Prompt Builder and LLM Ensemble.** The prompt for LLM would be built with these components: the meta-information describing the task and relevant insights, one or some prior programs the current parent program to be improved, the evaluation scores of these programs, and final instructions including the code-replacement rules, etc. Here the programs are sampled from program database. Given the prompt, LLM ensemble would generate a response with reasoning CoT and one or more `SEARCH/REPLACE` diff blocks that modify the parent program.

**Verifier.** The LLM response is then processed by the parser to extract diff blocks, which are applied to the parent program to obtain a child program. This child program is subsequently evaluated using the task-specific evaluator. AlphaEvolve uses an asynchronous pipeline to enable parallel evaluation, as the evaluator often becomes the computational bottleneck due to its potentially large timeout (e.g., AlphaEvolve-v2 sets a 1000-second timeout for the `FirstAutoCorrIneq` problem). Finally, the child program and its evaluation result are added to the database,

where they are reranked and organized as described earlier.

## 3. ThetaEvolve Key Features

In this section, we introduce the key features of ThetaEvolve. We mention the most important features, and leave other details in Appendix C.

### 3.1. Direct Adjustment

First, we make several straightforward simplifications or modifications relative to AlphaEvolve/OpenEvolve. They includes: (1) **Single LLM.** Unlike previous works that emphasize LLM ensembles, we only use a single LLM in ThetaEvolve. (2) **Large Program Database.** We use a much larger program database (`population_size = 10000`) in ThetaEvolve compared with OpenEvolve (`population_size` = 70; AlphaEvolve does not specify the exact size). As shown in Sec. 4.4.1, *scaling the database size improves final performance as the test-time compute increases*. **(3) (Optional) Iterative Refinement.** We sample only the parent program, without including additional prior programs as in AlphaEvolve, resulting in a simplified iterative refinement procedure. It is optional and is used primarily to reduce the prompt length.

### 3.2. Batch Sampling and Generation

In AlphaEvolve, each iteration builds only a single prompt and obtains one LLM response. Although it uses an asynchronous pipeline, it is still not efficient enough when scaling test-time compute, as it cannot fully leverage optimized batched inference engines such as vLLM (Kwon et al., 2023) or SGLang (Zheng et al., 2024). Therefore, we generate multiple responses from a batch of different parent programs to improve the inference efficiency. As shown in the bottom of Fig. 1, at each step, ThetaEvolve independently samples $B$ parent programs from the database, producing $B$ prompts. Then, $n$ responses are generated for each prompt, yielding a total of $B \times n$ child programs. These responses and their metrics can also be used for RL training. When inserting these programs into the database, we add them sequentially and re-organize the database after each insertion, which incurs negligible overhead compared with other system operations.

### 3.3. Early Check and Lazy Penalty

Since the LLM may generate responses with various issues, such as missing `SEARCH`/`REPLACE` diff blocks or producing child code with compilation errors, we perform a series of early checks to avoid unnecessary evaluation and assign penalty scores ($< 0$) to such cases. In detail, given a parent program $pp$, an LLM response $r$, a child program $cp = cp(pp, r)$ produced by the parser, and an evaluator function $\mathcal{E}$, we apply the following checks:

$$s(pp, r) = \begin{cases} -0.4, & \textit{if} \text{ no diff blocks are found in } r, \\ -0.3, & \textit{elif} \text{ no valid changes } (cp \equiv pp), \\ -0.2, & \textit{elif} \text{ no solution}, \\ -0.1, & \textit{elif} \text{ invalid solution}, \\ \mathcal{E}(cp), & \text{otherwise}. \end{cases} \tag{1}$$

Here, "no solution" includes different kinds of cases that prevent the program from getting any solution, like compilation errors, execution errors, and timeouts. "Invalid solution" means the program successfully produces an solution, but it fails the evaluator's validity checks (e.g., overlapping circles in `CirclePacking`).

Notably, we penalize all non-valid changes, i.e., any child program $cp$ that is equivalent to its parent program $pp$ up to comment removal. This is helpful for RL training: because improving solutions to open mathematical problems is difficult, most modifications do not yield better scores, especially when performance is near the best-known results.

### 3.4. (Optional) RL Reward Shaping

Our ThetaEvolve system provides an adaptive verifiable environment (Zeng et al., 2025) for RL training. A natural choice for the reward is the original objective score, which works for tasks like `CirclePacking`. However, some tasks may have more narrow ranges of objective values (e.g., $0.90 \sim 0.96$ for `SecondAutoCorrIneq`) which can not effectively differentiate rewards for different solution. Therefore, we normalize the reward for some tasks to provide a more stable and effective training signal. Specifically, for a given objective score $s$ (possibly obtained from an early check), we define the reward function $\mathcal{R}$ as:

$$\mathcal{R}(s) = \begin{cases} s, & \textit{if } s < 0 \text{ or reward shaping is disabled}, \\ k \cdot \mathcal{F}(s), & \text{otherwise}. \end{cases} \tag{2}$$

Here, $\mathcal{F} : [0, \infty) \to [0, 1]$ is the reward-shaping function and $k \in \mathbb{R}^+$ is the scaling factor of reward, which is set to be 3 in our paper. For each task, we manually specify the upper and lower bounds of the objective value as $U$ and $L$, together with a factor $\alpha \geq 1$. We then define $\mathcal{F}$ as:

$$\mathcal{F}(s) = \{\text{clip}(\mathcal{H}(s), 0, 1)\}^{\alpha}, \tag{3}$$

where a larger $\alpha$ rewards higher scores more aggressively as the score approaches the best-known value, and $\mathcal{H}(s)$ is a simple linear mapping that transforms $[L, U]$ to $[0, 1]$:

$$\mathcal{H}(s) = \begin{cases} (s - L)/(U - L), & \textit{if} \text{ maximizing}, \\ (U - s)/(U - L), & \text{otherwise}. \end{cases} \tag{4}$$

Details of the reward-shaping parameters for each task are

provided in Appendix C.2, and we present ablation studies and recommended setup for reward shaping in Sec. F.7.

## 4. Experiments

In our experiments, we present new best-known bounds obtained by scaling with ThetaEvolve (Sec. 4.2), analyze RL training under ThetaEvolve (Sec. 4.3), and ablate key components of the framework (Sec. 4.4). We first introduce the experimental setup below.

### 4.1. Setup

**Model.** We consider two open-source small models: `ProRL-1.5B-v2` (Hu et al., 2025) and `DeepSeek-R1-0528-Qwen3-8B` (DeepSeek-AI, 2025) (denoted `Distill-Qwen3-8B`). They contain far fewer parameters than frontier models, but have competitive capabilities among models of similar scale.

**Pipeline.** We build our program-evolution dynamic environment based on OpenEvolve (Sharma, 2025), an open-source implementation of AlphaEvolve. We utilizes `slime` framework (Zhu et al., 2025) for RL training and batch inference. For RL, we choose GRPO (Shao et al., 2024) augmented with asymmetric clipping (Yu et al., 2025). For the main experiments, we use three random seeds (42, 1234, 3407) to reduce variance. We note that scaling compute by running with more random seeds may further improve final performance. More details are in Appendix C.1.

**Tasks.** We evaluate five open mathematical problems. Four of them originate from AlphaEvolve (Novikov et al., 2025): (1) `CirclePacking-T`: pack $N = 26$ circles into a unit square while maximizing the sum of radii; (2/3/4) `FirstAutoCorrIneq` / `SecondAutoCorrIneq` / `ThirdAutoCorrIneq`: improve the constant bounds for the first, second, and third autocorrelation inequalities by constructing specialized functions; and (5) `HadamardMatrix`: hadamard maximal determinant problem, maximize the determinant of a matrix with $N = 29$. `FirstAutoCorrIneq` and `ThirdAutoCorrIneq` are minimization problems, while the others are maximization ones. Task descriptions, meta information, and task-specific parameters are detailed in Appendix B, C.3, and C.2, respectively.

Importantly, we note that the evaluation setups in OpenEvolve and AlphaEvolve differ slightly for circle packing problems. OpenEvolve permits a tolerance of $1 \times 10^{-6}$ when checking whether circles overlap or lie outside the unit square (Appendix B.1). This minor difference leads to slightly different optimization problems. In our work, we follow OpenEvolve's evaluation setup (`CirclePacking-T`, **T**olerance), and refer to the strict

AlphaEvolve version as a separate task, `CirclePacking`. We still achieve new SOTA results on `CirclePacking` by simply slightly shrinking the radii of the solutions obtained for `CirclePacking-T`.

### 4.2. Main Result: Scaling with ThetaEvolve

Firstly, we present the main experiments with ThetaEvolve on two models and four open optimization problems, including both pure-inference and RL training runs. The results are shown in Tab. 2. Impressively, for `Distill-Qwen3-8B`, ThetaEvolve achieves better results on `CirclePacking` than AlphaEvolve in both the RL and no-RL settings. In Appendix F.5, we further show that our solution has a slightly different configuration from the AlphaEvolve solution: ours is asymmetric, whereas AlphaEvolve's is symmetric. We also note that our solution is visually close to the one found by ShinkaEvolve; however, ShinkaEvolve uses an ensemble of six frontier LLMs such as Claude-Sonnet-4, o4-mini, and GPT-4.1, and their program requires around 75 seconds to find the solution, while ours takes only about 3 seconds. See Appendix F.2 for the analysis of our program.

Across all tasks, we also observe that ThetaEvolve with RL consistently outperforms pure inference, even with fewer training steps (each corresponding to 512 new programs). Both settings significantly improve upon the initial programs. At first glance, the improvements may appear small, but it is important to emphasize that as a bound approaches the best-known value, further gains become much more difficult, and even small improvements are non-trivial. Moreover, solutions with similar scores can still differ meaningfully in structure. To illustrate this, we visualize several solutions in Fig. 2. Although some runs achieve close numerical scores, their constructed functions show clear qualitative differences. For example, on `ThirdAutoCorrIneq`, `ProRL-1.5B-v2` achieves a score around 1.6 while `Distill-Qwen3-8B` achieves around 1.5, seemingly a modest difference compared the difference with initial program's 3.1586, yet Fig. 2 reveals that the function constructed by `Distill-Qwen3-8B` is substantially more complex than that of `ProRL-1.5B-v2`.

### 4.3. Analysis of RL training

Furthermore, we analyze the effect of applying RL training with ThetaEvolve. Ablation studies about format reward and reward shaping are provided in Appendix F.6 and F.7.

#### 4.3.1. DOES THE MODEL REALLY LEARN TO EVOLVE?

To verify whether the RL process in ThetaEvolve helps the model learn useful evolutionary strategies, we visualize the training curve of `ProRL-1.5B-v2` on `CirclePacking-T` (abbreviated as "CP" in this section). The results are shown in Fig. 3, left. In addition to the w/ RL

*Table 2.* **Main results.** For each task, we compare w/ RL and w/o RL under different training steps for `ProRL-1.5B-v2` and `Distill-Qwen3-8B`. "↑" denotes maximization tasks and "↓" denotes minimization tasks. We report the mean and best scores across three seeds. More details are in Tab. 7, and related parameters are list in Tab. 6. Notably, the best result achieved by AlphaEvolve on `CirclePacking` is 2.63586276. Although our evaluator includes a $10^{-6}$ tolerance in the validity checks as in OpenEvolve, it is easy to prove that, even after uniformly shrinking the radii of all circles by $10^{-6}$, the solutions from our runs on `Distill-Qwen3-8B` remain better than that of AlphaEvolve.

| Task | Method | ProRL-1.5B-v2 (Hu et al., 2025) | | | Distill-Qwen3-8B (DeepSeek-AI, 2025) | | |
|---|---|---|---|---|---|---|---|
| | | Step | Mean | Best | Step | Mean | Best |
| CirclePacking-T (↑) | Initial | 0 | – | 0.9598 | 0 | – | 0.9598 |
| | w/ RL | 200 | **2.3498** | **2.5225** | 65 | **2.6359840** | **2.6359857** |
| | w/o RL (early) | 200 | 2.0265 | 2.1343 | 65 | 2.6354195 | 2.6359831 |
| | w/o RL (late) | 600 | 2.0991 | 2.2491 | 100 | 2.6359541 | 2.6359834 |
| ThirdAutoCorrIneq (↓) | Initial | 0 | – | 3.1586 | 0 | – | 3.1586 |
| | w/ RL | 200 | **1.6412** | **1.6053** | 65 | **1.5210** | **1.4930** |
| | w/o RL (early) | 200 | 1.6831 | 1.6155 | 65 | 1.5498 | 1.5084 |
| | w/o RL (late) | 600 | 1.6766 | 1.6123 | 100 | 1.5491 | 1.5084 |
| HadamardMatrix (↑) | Initial | 0 | – | 0.1433 | 0 | – | 0.1433 |
| | w/ RL | 100 | 0.4808 | **0.5635** | 65 | **0.5696** | **0.5764** |
| | w/o RL (early) | 100 | 0.3264 | 0.4961 | 65 | 0.5500 | 0.5733 |
| | w/o RL (late) | 300 | **0.4920** | 0.5375 | 100 | 0.5515 | 0.5733 |
| SecondAutoCorrIneq (↑) | Initial | | – | | 0 | – | 0.9055 |
| | w/ RL | | – | | 65 | **0.9444** | **0.9469** |
| | w/o RL (early) | | – | | 65 | 0.9411 | 0.9433 |
| | w/o RL (late) | | – | | 100 | 0.9418 | 0.9434 |

and w/o RL baselines, we include a third setting: we load the step-150 checkpoint from the best w/ RL run (whose best score is 2.5225), and then perform pure inference on top of this checkpoint (denoted as "Load_CP@150").

We observe that: (1) Consistent with the results in Tab. 2, w/ RL runs improve programs more quickly (in terms of training steps or number of generated responses/programs) than pure inference and also achieve better final performance. (2) Inference using the RL-trained checkpoint ("Load_CP@150") climbs even faster than the w/ RL runs and achieves a better best score than inference with the original model, though still slightly worse than the full RL run. This indicates that RL meaningfully updates the model parameters in ways that benefit program evolution.

Moreover, we evaluate this CirclePacking-trained checkpoint on unseen tasks, as shown in Fig. 3, middle and right. We find that, compared to the base model, this checkpoint significantly improves average performance, often matching or even surpassing the w/ RL runs on those tasks, and slightly improves the best performance as well. This suggests that RL with the ThetaEvolve dynamic environment may enable the model to acquire an evolution capability that transfers across tasks, providing a positive signal that this single-task RL training paradigm could potentially be extended into a more general post-training recipe.

### 4.3.2. COMPARISON WITH VANILLA RL TRAINING

We further highlight the importance of applying RL with a *dynamic* environment that stores and updates experience, as in ThetaEvolve. We compare our results with a baseline that applies RL in a static environment, i.e., always starting from the initial program, which is also used in AlphaEvolve's ablation. The results, shown in Fig. 4, indicate a substantial performance gap: RL with a static environment performs much worse than RL with ThetaEvolve, and even worse than the pure inference baseline with ThetaEvolve. In Appendix E, we roughly analyze why this occurs: for challenging open problems, directly sampling the final advanced program is extremely unlikely. Thus, the task must be decomposed into a trajectory of incremental improvements, enabling the model to learn and operate at the frontier of its current capabilities.

### 4.4. Additional Analysis

In this section, we present ablation studies on key components of ThetaEvolve (database size, batch sampling), showing the effectiveness of our designs. In Appendix F.8, we also show that the database management strategies in AlphaEvolve/OpenEvolve remain important.

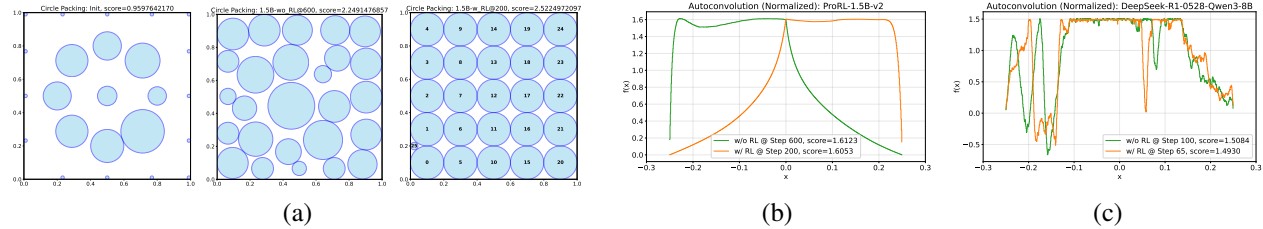

(a)                                    (b)                                    (c)

*Figure 2.* **Visualization of the solutions.** (a) `ProRL-1.5B-v2` on `CirclePacking-T`. (b) `ProRL-1.5B-v2` on `ThirdAutoCorrIneq`. (c) `Distill-Qwen3-8B` on `ThirdAutoCorrIneq`. Although the scores of the w/ RL and w/o RL solutions are close, the solutions themselves (and the corresponding programs) may differ noticeably. We normalize the functions (see Appendix B) for clearer visualization. Besides, the constructed function obtained from `Distill-Qwen3-8B` is still much more complex than the one from `ProRL-1.5B-v2` (b and c), even though both are evolved with the same initial program and prompt.

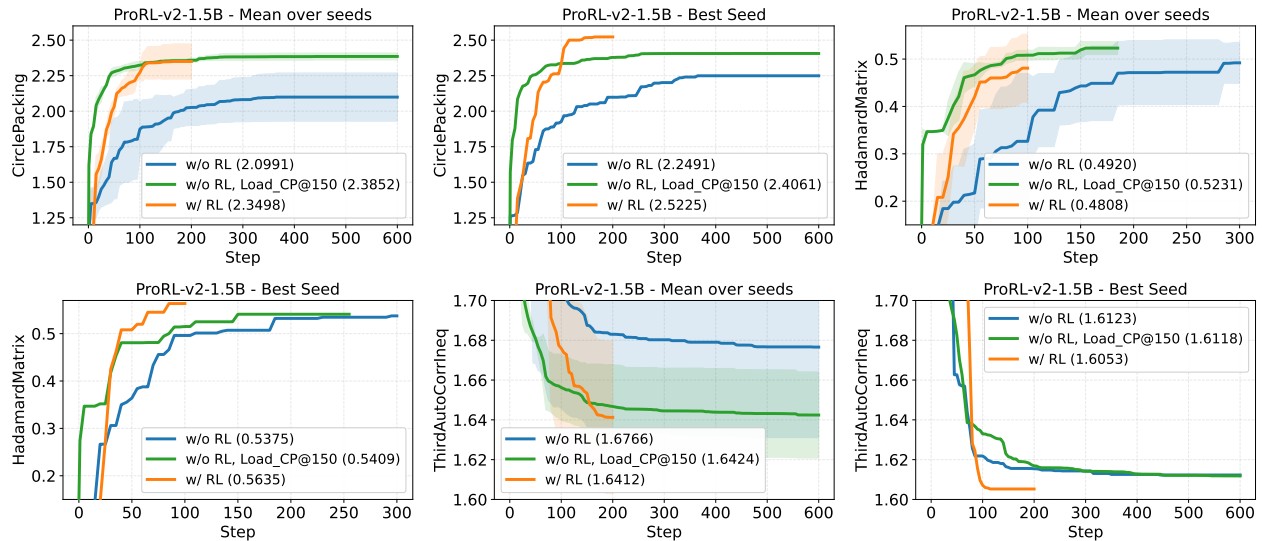

*Figure 3.* **RL-trained models outperform the base model in pure inference, both on the trained target task and on unseen tasks**. (1) w/o RL (blue): ThetaEvolve with inference-only on the original base model., (2) w/ RL (orange): ThetaEvolve with RL training starting from the same base model. (3) w/o RL, Load_CP@150 (green): inference-only again, but using the step-150 checkpoint from the best RL run on CirclePacking, rather than the original base model. Shaded regions indicate standard deviation across different seeds.

|  | **Small** | **Medium** | **Large** |
|---|---|---|---|
| `population_size` | 70 | 1000 | 10000 |
| `archive_size` | 25 | 100 | 1000 |
| `num_islands` | 5 | 10 | 10 |

*Table 3.* Configurations for ablation study of database size. The **Small** setup is similar to that used in OpenEvolve.

| **Pipeline** | **#Programs** | **Mean** | **Best** |
|---|---|---|---|
| AlphaEvolve | | 2.6358628 | |
| OpenEvolve | 512 | 1.0955 | 1.2634 |
| OpenEvolve | 307.2k | 2.1313 | 2.1773 |
| **ThetaEvolve w/o RL** | 307.2k | 2.0991 | 2.2491 |
| **ThetaEvolve w/ RL** | 307.2k | **2.3498** | **2.5225** |

*Table 4.* **Ablation study across different pipelines.** Model: `ProRL-1.5B-v2`, task: `CirclePacking-T`.

| **Pipeline** | **Time (h)** |
|---|---|
| OpenEvolve | 63.6 |
| **ThetaEvolve (w/o RL)** | **5.4** |

*Table 5.* **Speed comparison for different pipelines.** Here we run `ProRL-1.5B-v2` on `CirclePacking-T` for 400 inference steps. Experiments are conducted on 4 A6000.

### 4.4.1. SCALING DATABASE FOR COLLABORATING WITH SCALING TEST-TIME COMPUTE

Firstly, we show that *scaling the size of the program database is important when increasing test-time compute*. Notably, OpenEvolve include three key parameters related to database size: (1) `population_size`: the maximum number of programs that can be stored in the database; (2) `archive_size`: the size of the elite archive, from which programs are sampled with higher probability for exploitation; (3) `num_islands`: the number of independent-subgroups in evolution, in general the larger the more diver-

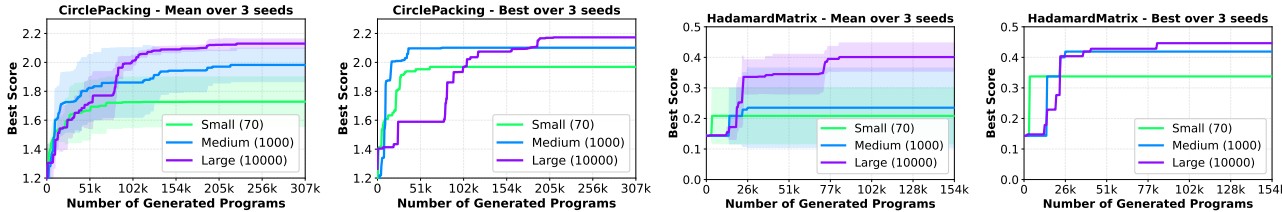

*Figure 4.* **RL with ThetaEvolve dynamic environment outperform RL with static environment.** Here static environment means always starting with initial program, similar to the ablation baseline used in AlphaEvolve (Novikov et al., 2025).

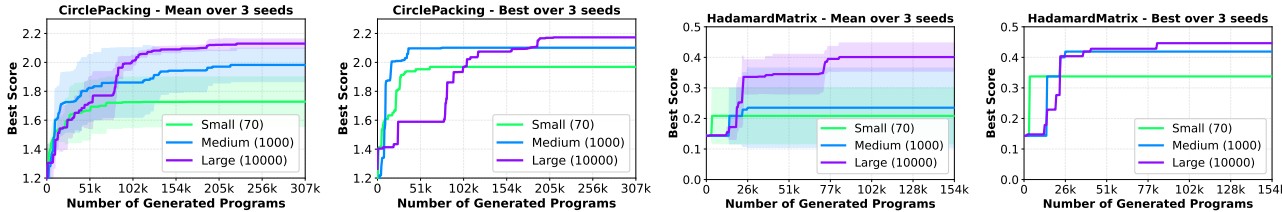

*Figure 5.* **Scaling database size improves the performance of program evolution, especially when increasing test-time compute.** Here we consider `ProRL-1.5B-v2`, plot the mean and highest values of the best objective score across evolution runs with 3 seeds. (**Left**) Evaluate `CirclePacking-T`, using the default OpenEvolve pipeline (Fig. 1, top) except that we vary the database size as described in Sec. 4.4.1. (**Right**) Similar OpenEvolve setup for `HadamardMatrix`.

sity. (Check Appendix D.4 for detailed illustration). We set the database-size configurations as listed in Tab. 3, and the results of ablating database size are presented in Fig. 5.

We see that when test-time compute is relatively small (e.g., fewer than 40 inference steps or fewer than 20K generated programs), a smaller database can progress faster because high-scoring programs are sampled more frequently (Note that the *Large* database requires approximately 10K programs (or roughly 20 inference steps) before it becomes fully populated and begins discarding low-scoring programs). However, when further scaling test-time compute, it always have very limited additional improvement, while increasing the database size improves the diversity of candidate programs, which in turn strengthens the effectiveness of the evolutionary search.

### 4.4.2. BATCH SAMPLING

In this section, we scale the test-time compute in the original OpenEvolve pipeline, which uses asynchronous sequential sampling, and compare it with ThetaEvolve. We show that ThetaEvolve without RL can perform as well as OpenEvolve, even though its program database is updated in a less online manner due to batch-based program generation, while achieving significantly faster inference. Here, we serve `ProRL-1.5B-v2` using vanilla `SGLang` (Zheng et al., 2024) with the same inference parameters (e.g., TP, dtype) used in ThetaEvolve. The results are shown in Tab. 4.

We observe that when generating only a small number of new programs (e.g., ∼500), similar to OpenEvolve's default setup, `ProRL-1.5B-v2` exhibits very limited improvement over the initial program. However, when scaling the

test-time compute of OpenEvolve to match ThetaEvolve w/o RL (307.2k new programs), the OpenEvolve pipeline also achieves a similarly large improvement, though it still underperforms RL-trained runs. This highlights that *scaling test-time compute is essential for evolving tasks, regardless of the inference pipeline*. In addition, inference with ThetaEvolve is much faster than with OpenEvolve, as shown in Tab. 5. We attribute this to batch sampling, which provides much higher throughput for the inference engine compared to asynchronous sequential requests.

### 4.4.3. MORE COMPARISON WITH ALPHAEVOLVE

Finally, note that the second version of the AlphaEvolve report (Georgiev et al., 2025) provides an optional initial program and prompt for `FirstAutoCorrIneq`, it allows us to directly compare our setup with AlphaEvolve. We consider two setups and the results are visualized in Fig. 6.

(1) We use the provided initial program (with score 1.5214) and prompt from AlphaEvolve-v2 (Fig. 6, Left), and keep the initial program start from random solution as AlphaEvolve. We observe that after around 50 steps, `Distill-Qwen3-8B` discovers a step function whose auto-convolution is very similar to that found by AlphaEvolve-v2 (Georgiev et al., 2025) (Fig. 6 Middle). Although our objective score (1.5068) is still worse than the SOTA value (1.5032), it already surpasses the previous human SOTA result (1.5097) (Matolcsi & Vinuesa, 2010). A similar pattern appears in `SecondAutoCorrIneq`: although our score (0.9469) is worse than the most recent AlphaEvolve-v2 result (0.9610), it is still better than the previous human best result (0.9414) (Jaech & Joseph, 2025a). Notably, the timeout of our program is only 350 seconds,

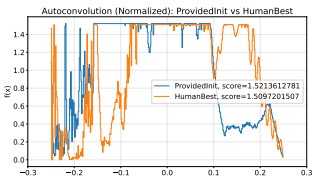 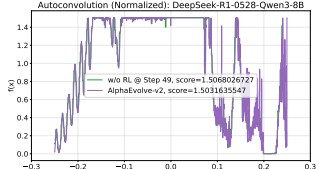 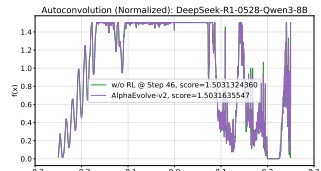

*Figure 6.* **Comparison with AlphaEvolve under the same initial program and prompt.** We consider `FirstAutoCorrIneq`. (**Left**) The solutions and scores of the previous human SOTA (Matolcsi & Vinuesa, 2010) and the initial program provided in AlphaEvolve-v2. (**Middle**) Running ThetaEvolve without RL, we find that although our score is worse than the new best-known function found in AlphaEvolve-v2, our evolved program produces an autoconvolution curve that is highly similar, and our solution still outperforms the previous human SOTA. (**Right**) When we additionally initialize with the AlphaEvolve-v2 SOTA solution, ThetaEvolve can make slight further improvements to the bound.

whereas the programs used for the human best bound or AlphaEvolve-v2 often require several hours.

(2) Beyond (1), we further use the AlphaEvolve-v2 SOTA solution as an additional initialization. In this setting, `Distill-Qwen3-8B` can still make slight improvements over the SOTA solution. Although the generated programs primarily apply small perturbations to the step function, improving local optimization efficiency but not exploring more aggressive modifications as AlphaEvolve-v2, we mention that this initialization may already be a local minimum that hard to further improve from.

## 5. Discussion on Future Work

In general, AlphaEvolve and ThetaEvolve are broad pipelines suitable for optimization problem with a continuous reward, making them applicable to a wide range of real-world tasks beyond open mathematical problems. Moreover, the task-transfer phenomenon observed in Sec. 4.3.1 suggests that we may be able to train on multiple targets simultaneously, for example, using different instances of the same task with varying parameters or even combining entirely different tasks. This could potentially extend to post-training workflows as well. Finally, we emphasize that enabling a model to continually learn may require replacing a static environment with a dynamic one that co-evolves with the model, which may also provide insights for effective exploration strategy in RL training. Our work offers an early attempt at applying RL to a dynamic, verifiable environment controlled by a context manager (such as a program database), and we believe there is substantial room for further improvement and optimization.

## Acknowledgements

We thank Liyuan Liu, Lifan Yuan, Pang Wei Koh, Jerry Li, Gregory Lau, Rulin Shao and Jingming Zhuo for very helpful discussions. YW, ZZ, and XY are supported by Amazon AI Ph.D. Fellowships. SSD acknowledges the support of NSF CCF-2212261, NSF IIS-2143493, NSF CCF-2019844, NSF IIS-2229881, the Sloan Research Fellowship, and Schmidt Sciences AI 2050 Fellowship.

## Impact Statement

This paper presents work whose goal is to advance the field of Reinforcement Learning and AI for Math. There are many potential societal consequences of our work, none of which we feel must be specifically highlighted here.

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

## A. Related Work

Previous work has incorporated LLMs into the evaluation loop for prompt optimization, where the model iteratively updates contextual information in the prompt based on feedback to improve downstream performance (Yang et al., 2023; Khattab et al., 2023; Fernando et al., 2023; Guo et al., 2023; Madaan et al., 2023; Agrawal et al., 2025; Surina et al., 2025). A related line of work on agentic LLMs maintains trajectory information or feedback in explicit context managers (Shinn et al., 2023; Zhang et al.), and then surfaces this experience in subsequent prompts, and some recent works also include test-time training for agentic flows (Zhang et al., 2025; He et al., 2025a;b), though their scenarios are different from optimizing open problems.

By contrast, recent pipelines such as FunSearch (Romera-Paredes et al., 2024), AlphaEvolve (Novikov et al., 2025; Georgiev et al., 2025) and CodeEvolve (Assumpção et al., 2025) focuses on more specific goals for in-context evolving, e.g., program optimization for continuous objectives on challenging open problems. Recent work has also studied the co-evolution of algorithms and language models for automatic heuristic design (Huang et al., 2025). While related in its use of parameter updates during program evolution, this line of work mainly targets standard optimization benchmarks with training and test instances. In contrast, ThetaEvolve focus on test-time learning for a single open optimization problem without training sets, and supports AlphaEvolve-style full-program modification over a broader search space. Other evolutionary LLM systems emphasize sample efficiency through mechanisms such as reflection (Ye et al., 2024), which is complementary to our focus on scaling test-time compute for cheaply verifiable open problems. These prompt-optimization and evolutionary program-search systems are still predominantly inference-time pipelines, so the underlying LLM does not fully internalize the discovered capabilities.

On the other hand, AlphaProof (Hubert et al., 2025) couples a pre-trained LLM with an AlphaZero-style (Silver et al., 2018) reinforcement learning loop in the Lean proof assistant (de Moura et al., 2015), which serves as a self-contained automated verifier. Beyond its large-scale offline RL training, AlphaProof further employs Test-Time RL (TTRL): at inference time, it generates a curriculum of formal variants around a hard target problem and continues RL training on these variants within the Lean environment, enabling strong problem-specific adaptation that substantially boosts its formal proving performance. Motivated by these directions, ThetaEvolve treats an AlphaEvolve-style program evolution pipeline as an adaptive verifiable environment (Zeng et al., 2025; Shao et al., 2025a), and applies RL to optimize programs for continuous-reward objectives.

## B. Details of Tasks

In this section, we describe the details of the tasks evaluated in our paper.

### B.1. Circle Packing

`CirclePacking` is defined as follows: given a positive integer $n$, the task is to pack $n$ disjoint circles inside a unit square so as to maximize the sum of their radii.

For the case $n = 26$, the previous best-known value was 2.634 (Friedman, 2012), and AlphaEvolve (Novikov et al., 2025) improved this result to 2.63586276. More recently, ShinkaEvolve (Lange et al., 2025) further increased the best-known value to 2.635983283.

One implementation detail worth noting is that different pipelines adopt slightly different numerical tolerances when checking the configuration. For example, in the official OpenEvolve implementation, the evaluation function uses an absolute tolerance of $\text{atol} = 10^{-6}$ when checking the constraints. ShinkaEvolve adopts a similar approach with $\text{atol} = 10^{-7}$, whereas AlphaEvolve uses zero tolerance for overlap detection. Because of this discrepancy, ShinkaEvolve reports two values for the packing: one evaluated with $\text{atol} = 10^{-7}$ and one with $\text{atol} = 0$. In our experiments, we adopt the OpenEvolve-style evaluation (`CirclePacking-T`), and consider the setting $\text{atol} = 0$ as the formal task `CirclePacking`. We can simply shrink the radii of circles to obtain the results for `CirclePacking` by the program found on `CirclePacking-T`.

### B.2. First Autocorrelation Inequality

For a function $f : \mathbb{R} \to \mathbb{R}$, the autoconvolution of $f$ is given by

$$(f * f)(t) = \int_{\mathbb{R}} f(t - x) \, f(x) \, dx.$$

Define $C_1$ as the largest constant such that

$$\max_{-1/2 \leq t \leq 1/2} (f * f)(t) \geq C_1 \left( \int_{-1/4}^{1/4} f(x)\, dx \right)^2$$

holds for all non-negative functions $f : \mathbb{R} \to \mathbb{R}$. This inequality is closely connected to additive combinatorics, particularly questions concerning the size of Sidon sets. The best-known bounds satisfy

$$1.28 \leq C_1 \leq 1.5098,$$

where the lower bound was established previously (Cloninger & Steinerberger, 2017), and the upper bound originates from a step-function construction (Matolcsi & Vinuesa, 2010). AlphaEvolve-v2 (Georgiev et al., 2025) constructed a step function with 600 evenly spaced intervals over $[-1/4, 1/4]$, yielding the improved upper estimate

$$C_1 \leq 1.5032.$$

### B.3. Second Autocorrelation Inequality

Let $C_2$ denote the smallest constant such that

$$\|f * f\|_2^2 \leq C_2 \|f * f\|_1 \|f * f\|_\infty$$

holds for every non-negative function $f : \mathbb{R} \to \mathbb{R}$. Previously, the best lower bound for $C_2$ was obtained using a step-function construction (Matolcsi & Vinuesa, 2010), and AlphaEvolve-v1 further found a 50-piece step function that achieved a slightly improved lower bound of $0.8962$. Independently, another work established a stronger lower bound using gradient-based methods (Boyer & Li, 2025), obtaining

$$0.901564 \leq C_2 \leq 1,$$

and recent work further improved this bound by constructing a 2399-step function (Jaech & Joseph, 2025b), yielding

$$0.9414 \leq C_2 \leq 1.$$

Most recently, AlphaEvolve-v2 identified a step function with 50,000 pieces, raising the lower bound to

$$0.961 \leq C_2.$$

Notably, AlphaEvolve-v2 remarks that this function is highly irregular, both challenging to optimize and difficult to visualize, and is expected to yield an even higher score if the search budget is increased further.

### B.4. Third Autocorrelation Inequality

Let $C_3$ denote the largest constant for which

$$\max_{-1/2 \leq t \leq 1/2} |f * f(t)| \geq C_3 \left( \int_{-1/4}^{1/4} f(x)\, dx \right)^2$$

holds for every function $f : \mathbb{R} \to \mathbb{R}$. A step-function construction shows that

$$C_3 \leq 1.4581 \quad \text{(Cilleruelo et al., 2010)}.$$

AlphaEvolve identified a step function with 400 uniformly spaced intervals on $[-1/4, 1/4]$, yielding a slightly improved upper bound

$$C_3 \leq 1.4557.$$

However, there is a mismatch between the mathematical problem statement and the initial code implementation.

In the AlphaEvolve implementation, a step function is discretized as a height sequence $\{h_i\}_{i=1}^{n}$, and its autoconvolution is evaluated at $n$ discrete points $\{\text{conv}_3(k)\}_{k=1}^{n}$. The upper bound computed in AlphaEvolve is

$$C_3^{(\text{AlphaEvolve})} = \left| \frac{2n \, \max_k \text{conv}_3(k)}{\left( \sum_{i=1}^{n} h_i \right)^2} \right|.$$

If one discretizes the theoretical inequality in the natural way, the verification formula should instead be

$$C_3^{(\text{theory})} = \frac{2n \cdot \max_k |\operatorname{conv}_3(k)|}{\left(\sum_{i=1}^n h_i\right)^2}.$$

Clearly, $C_3^{(\text{theory})} \geq C_3^{(\text{AlphaEvolve})}$. Using this theoretically correct expression, neither the previously reported bound $1.4581$ nor the improved value $1.4557$ can be recovered by evaluating their corresponding step functions, their $C_3^{(\text{theory})}$ scores are substantially higher. This discrepancy reflects that two closely related but distinct optimization problems are being considered. In our paper, we use formula $C_3^{(\text{theory})}$ in our evaluator; therefore, our results are not directly comparable to the previously reported scores $1.4581$ or $1.4557$ in AlphaEvolve.

### B.5. Hadamard matrix

A Hadamard matrix is an $n \times n$ matrix $H$ with entries $\pm 1$ such that $HH^\mathsf{T} = nI$, where $I$ is the identity matrix. The maximal determinant problem asks for the largest possible value of $|\det(H)|$ over all $\{\pm 1\}$ matrices of order $n$, subject to the classical upper bound

$$|\det(H)| \leq n^{n/2}.$$

For $n = 29$, the best-known result was obtained by Orrick (Orrick et al., 2003), achieving

$$|\det(H)| = 2^{28} \times 7^{12} \times 320,$$

Following previous work, in our paper we compute the objective score for $n = 29$ as

$$s = \frac{|\det(H)|}{2^{28} \times 7^{12} \times 342}.$$

## C. Details of ThetaEvolve

We discuss additional implementation details of ThetaEvolve below.

### C.1. More Implementation Details

**Pipeline.** We set the batch size to $B = 32$ and the number of responses per prompt to $n = 16$ for both RL and pure inference in ThetaEvolve. The maximum response length is 16,384 tokens, and the rollout temperature is 1.0. We allow at most 16 verifier programs to run simultaneously.

**RL Training.** We use GRPO (Shao et al., 2024) as our RL algorithm, augmented with asymmetric clipping (Yu et al., 2025) using `clip_low = 0.2` and `clip_high = 0.28`. The learning rate is set to $10^{-6}$ and the weight decay to $0.1$. We additionally apply truncated importance sampling (Yao et al., 2025; Liu et al., 2025b;a) to improve training stability. Importantly, we do not incorporate dynamic sampling (Yu et al., 2025) to maintain a fair comparison between the number of samples used during RL training and inference. Nevertheless, dynamic sampling may further improve training stability and overall performance. We do not include KL-divergence or entropy regularization.

**Verifier.** For each problem, we provide two evaluator helper functions: one for validity checking and another for computing the objective score. We also require each program to save its solution in a separate file that is accessible to the evaluator defined in an immutable file. This design prevents reward hacking, such as attempts to modify the evaluation logic within the program file.

**Lazy Penalty.** In the main experiments, we simply penalize repetitions of the parent program. However, we optionally allow penalizing any child program that is equivalent to an existing program in the database, since the model may otherwise memorize and repeatedly output the best program stored there.

**Prompt Cleaning.** We also clean and unify some prompt templates from OpenEvolve, such as moving the metrics to the top of each program as done in AlphaEvolve.

**Mixing Meta-Information.** OpenEvolve introduces a two-stage setup that achieves performance close to AlphaEvolve on `CirclePacking-T`: (1) the system first guides the program to discover strong constructive solutions, and then (2) let the model optimizes a search algorithm that may use the previously found construction as an initialization trick. In ThetaEvolve, we simplify this process by allowing multiple pieces of meta information (kept fixed and not evolved as in AlphaEvolve), each describing a different strategy and associated with a corresponding weight. During generation, the prompt is constructed by sampling from these meta-information entries according to their weights.

Our experiments can be conducted on 8x80G A100s.

### C.2. Parameters in ThetaEvolve

The parameters used in ThetaEvolve are shown in Tab. 6. For reward shaping, we mainly apply it to tasks that have a narrow range of scores (`SecondAutoCorrIneq`) or are minimization tasks (`ThirdAutoCorrIneq`).

Notably, the value $L = 1.4557$ used for `ThirdAutoCorrIneq` is the best bound reported in AlphaEvolve (Novikov et al., 2025). However, as mentioned in Appendix B, it is actually the bound for $C_3^{\text{(AlphaEvolve)}}$ rather than $C_3^{\text{(theory)}}$. But since $C_3^{\text{(AlphaEvolve)}} \leq C_3^{\text{(theory)}}$, it is still reasonable to use it as the truncated lower bound $L$. We use $\alpha = 3$ for `Distill-Qwen3-8B`, since it can more easily achieve scores close to lower bound (e.g., 1.4930 from the best w/ RL run), but adopt a more conservative setup $\alpha = 1$ for `ProRL-1.5B-v2`. See the discussion in Sec. F.7.

| Model | Task | Timeout (s) | Reward Shaping? | $U$ | $L$ | $\alpha$ |
|---|---|---|---|---|---|---|
| `ProRL-1.5B-v2` | `CirclePacking-T` ($\uparrow$) | 70 | ✗ | - | - | - |
| `ProRL-1.5B-v2` | `ThirdAutoCorrIneq` ($\downarrow$) | 70 | ✓ | 2.5 | 1.5 | 1.0 |
| `ProRL-1.5B-v2` | `HadamardMatrix` ($\uparrow$) | 350 | ✗ | - | - | - |
| `Distill-Qwen3-8B` | `CirclePacking-T` ($\uparrow$) | 70 | ✗ | - | - | - |
| `Distill-Qwen3-8B` | `ThirdAutoCorrIneq` ($\downarrow$) | 70 | ✓ | 3.2 | 1.4557 | 3.0 |
| `Distill-Qwen3-8B` | `HadamardMatrix` ($\uparrow$) | 350 | ✗ | - | - | - |
| `Distill-Qwen3-8B` | `SecondAutoCorrIneq` ($\uparrow$) | 350 | ✓ | 0.96 | 0.91 | 1.0 |
| `Distill-Qwen3-8B` | `FirstAutoCorrIneq` ($\downarrow$) | 1150 | No RL runs | - | - | - |

*Table 6.* **Configurations of the main experiments.** Here the timeout is for the evaluator in OpenEvolve, which is slightly longer than the timeout specified in the prompt.

### C.3. Details of Meta Information

As mentioned in Appendix C.1, in ThetaEvolve, we sample the meta-information from several candidates. For `CirclePacking-T`, we use meta-information and an initial program that closely match those used in OpenEvolve. For `FirstAutoCorrIneq`, AlphaEvolve-v2 (Georgiev et al., 2025) provides both the meta-information and the initial program, and we adopt them directly for a fair comparison. For the remaining three tasks that do not have ready-made components, we prompt GPT-5 (OpenAI, 2025) and Claude 4.5 (Anthropic, 2025) to generate insights for improvement or to summarize ideas from previous work, following the emphasis in AlphaEvolve-v2 that verbal insights can significantly influence the evolution process, even for strong closed-source models. Importantly, we only use these models to generate insights and potential improvement directions for inclusion in the prompt, rather than directly producing advanced programs. All runs in our paper use the same prompt for each task to ensure a fair comparison. Almost all initial programs are relatively simple and achieve scores far away the best-known bounds. More details are illustrated below.

#### C.3.1. PROMPT OF CIRCLEPACKING-T

Notably, the prompts for `CirclePacking-T` are adapted from OpenEvolve's two-stage configuration. We preserve the main structure of the original prompts and only introduce minor modifications, such as adding information about timeouts and brief instructions encouraging the model to explore more. The prompt contains two components (Here we set: `n_circles = 26`, `MAX_RUNTIME = 60`, `target_value = 2.635`):

(1) Part 1, weight = 0.3:

```
You are an expert mathematician specializing in circle packing problems and
computational geometry.
Your task is to improve a constructor function that directly produces a specific
```

```
arrangement of {core_parameters.n_circles} circles in a unit square,
maximizing the sum of their radii. The AlphaEvolve paper achieved a sum of
{target_value} for n={core_parameters.n_circles}.
The time limit for each program evaluation is {MAX_RUNTIME} seconds.

Key geometric insights:
- Circle packings often follow hexagonal patterns in the densest regions
- Maximum density for infinite circle packing is pi/(2*sqrt(3)) approx 0.9069
- Edge effects make square container packing harder than infinite packing
- Circles can be placed in layers or shells when confined to a square
- Similar radius circles often form regular patterns, while varied radii allow better
space utilization
- Perfect symmetry may not yield the optimal packing due to edge effects

Focus on designing an explicit constructor that places each circle in a specific
position, rather than an iterative search algorithm.
```

(2) Part 2, weight = 0.7:

```
You are an expert mathematician specializing in circle packing problems and
computational geometry. We're trying to reach the AlphaEvolve target of {target_value}
for the sum of radii when packing {core_parameters.n_circles} circles in a unit square.
The current implementation has plateaued at some values, so we need significant
improvements.
The time limit for each program evaluation is {MAX_RUNTIME} seconds.

Key insights to explore:
1. The optimal arrangement likely involves variable-sized circles
2. A pure hexagonal arrangement may not be optimal due to edge effects
3. The densest known circle packings often use a hybrid approach
4. The optimization routine is critically important - simple physics-based models with
carefully tuned parameters
5. Consider strategic placement of circles at square corners and edges
6. Adjusting the pattern to place larger circles at the center and smaller at the edges
7. The math literature suggests special arrangements for specific values of n
8. scipy has some useful functions for optimization

Focus on breaking through the plateau by trying fundamentally different approaches -
don't just tweak parameters.

IMPORTANT: If you find the previous programs produce similar results, try as creative
and evolutionary strategies as possible to explore different approaches.
```

### C.3.2. PROMPT OF FIRSTAUTOCORRINEQ

The prompts for FirstAutoCorrIneq are adapted from the one released by AlphaEvolve-v2 (Georgiev et al., 2025).
We preserve the most of the parts of the original prompts (here we set: MAX_RUNTIME = 1000):

(1) Part 1, weight = 1.0:

```
You are an expert mathematician and computational scientist specializing in harmonic
analysis and extremal problems, specifically the first autocorrelation inequality.
Your task is to generate the sequence of non-negative heights of a step functions on
the domain {core_parameters.domain}, that minimizes the following evaluation function:

def evaluate_sequence_1(sequence: list[float]) -> float:
  """Evaluates a sequence of coefficients."""

  # Protect against negative numbers
  sequence = [max(0.0, x) for x in sequence]
  n = len(sequence)
  b_sequence = np.convolve(sequence, sequence)
  max_b = max(b_sequence)
  sum_a = np.sum(sequence)
```

```
  return float(2 * n * max_b / (sum_a**2))
```

You don't have to change the evaluation function in the program, it would be provided
in the evaluation environment that you cannot modify.

Your task is to write a search function that searches for the best sequence of
coefficients.
Your function will have {MAX_RUNTIME} seconds to run, and after that it has to have
returned the best sequence it found.
If after {MAX_RUNTIME} seconds it has not returned anything, it will be terminated
with negative infinity points.
All numbers in your sequence have to be positive or zero. You may code up any search
method you want.

Feel free to change parameters like the length of the sequence, or any other
parameters you deem necessary to improve performance.

### C.3.3. PROMPT OF SECONDAUTOCORRINEQ

We crafted our prompt by drawing on insights from two papers that achieved state-of-the-art results in this domain (Boyer
& Li, 2025; Matolcsi & Vinuesa, 2010). The prompt contains three components (here we set: domain = [-1/4, 1/4],
MAX_RUNTIME = 300, target_value = 0.96):

(1) Part 1, weight = 0.3:

```
You are an expert in functional optimization and harmonic analysis on autoconvolution
inequalities.
Your task is to explicitly construct a single non-negative function f on
{core_parameters.domain} to maximize

R(f) = ||f * f||_2^2 / (||f * f||_1 * ||f * f||_inf), with C_2 >= R(f) and target R(f)
> {core_parameters.target_value}.
The time limit for each program evaluation is {MAX_RUNTIME} seconds.

High-impact constructive insights (no iterative search here):
- Use a two-/multi-scale piecewise-constant scaffold: an off-center tall narrow spike
riding on a broad, low-amplitude envelope. The goal is to inflate the L2 mass of f*f
while flattening its global peak.
- Add a phase-locked micro-comb with spacing aligned to the dominant offsets of the
current envelope autocorrelation; keep teeth weak but well-phased to cancel incipient
maxima in f*f and broaden the central plateau.
- Allow asymmetry deliberately; biasing mass away from the center often widens the
plateau without raising ||f*f||_inf.
- Parameterize nonnegativity by construction (e.g., nonnegative coefficients of
bumps/steps) and keep a small number of tunable knobs (spike location/width/height,
envelope level, comb spacing/phase, local refinement windows) so the ansatz evaluates
instantly.
- Output a pure constructor that returns the heights array; do not implement any
search/loop in this prompt.

IMPORTANT - Summary: Provide a fast, high-resolution ansatz (off-center spike +
envelope + phase-locked weak comb) that raises ||f*f||_2 and suppresses ||f*f||_inf by
design, serving as a strong warm start for downstream optimizers.
```

(2) Part 2, weight = 0.4:

```
You are an expert in differentiable optimization for autoconvolution objectives.
We need a peak-aware, curvature-sensitive optimizer to push

R(f) = ||f * f||_2^2 / (||f * f||_1 * ||f * f||_inf) beyond
{core_parameters.target_value}.
The time limit for each program evaluation is {MAX_RUNTIME} seconds.

Optimization insights (concise, no step-by-step plan):
```

```
- Replace the hard max by a temperature-annealed Top-k smooth max (softmax over the
top few entries of f*f). This focuses gradients on the peak set, not on noise;
validate with the true max when reporting.
- Use Toeplitz-aware preconditioning: precondition updates by local convolutional
smoothing (e.g., with a short triangular kernel) to approximate the inverse curvature
of the f -> f*f map; this damps oscillatory directions that spuriously raise
||f*f||_inf.
- Adopt a mirror-descent / multiplicative parameterization (e.g., optimize a base
field then apply softplus/exponential), enforcing f >= 0 and naturally emphasizing
relative (scale-aware) updates in high-impact regions.
- Run multi-resolution refinement in-place: briefly expose local high-resolution
windows only near indices that control the Top-k of f*f, then fold improvements back
to the global grid. Prefer short decisive bursts over long loops.
- Inject structured, peak-targeted perturbations (small, phase-coherent comb nudges
around the current argmax offsets) to prevent peak lock-in while preserving the
plateau that boosts ||f*f||_2.

Diagnostics to log (lightweight): current R(f) with hard/smooth max, Top-k peak values
and gap, estimated plateau width.

IMPORTANT - Summary: Use Top-k smooth max, Toeplitz-aware preconditioning, mirror
descent, and local multi-resolution to lower the peak without shrinking the plateau,
enabling rapid gains within the short time budget.
```

(3) Part 3, weight = 0.3:

```
You are an expert in structural search for autoconvolution bounds under tight compute
budgets.
Goal: reveal diverse, high-payoff families of nonnegative f on
{core_parameters.domain} that lift

R(f) = ||f * f||_2^2 / (||f * f||_1 * ||f * f||_inf) above
{core_parameters.target_value}.
The time limit for each program evaluation is {MAX_RUNTIME} seconds.

High-signal exploration heuristics (keep them lightweight):
- Frequency shaping: design envelopes that suppress Fourier modes responsible for the
argmax of f*f; add notch filters via weak combs whose spacing is locked to the argmax
offsets.
- Asymmetric dual-spike with beat control: two offset spikes with height/spacing tuned
to create a broad, nearly flat plateau in f*f through controlled interference; keep
the secondary spike weaker to avoid a new global maximum.
- Adaptive resolution windows: vary total segment counts aggressively but allocate
fine resolution only near sensitivity hot zones (indices impacting Top-k peaks of
f*f); prefer fast trials over long runs.
- Budget-aware gating: use quick proxy signals (Top-k peak gap, plateau width
estimate) to promote only the most promising shapes for any expensive refinement,
avoiding uniform spend.

Rank/Report: R(f) (hard max), Top-k peak gap, plateau width proxy, and whether the
best peak index stabilizes under small perturbations.

IMPORTANT - Summary: This prompt prioritizes frequency-domain flattening, beat-pattern
plateaus, and targeted resolution to uncover novel high performers quickly, maximizing
diversity per minute.
```

### C.3.4. PROMPT OF THIRDAUTOCORRINEQ

We designed our prompt by drawing on insights from the thesis (Cilleruelo et al., 2010) that achieved state-of-the-art results in this domain, though we find that there are some mismatch in evaluation later (Appendix B). The prompt contains three components (here we set: domain = [-1/4, 1/4], MAX_RUNTIME = 60, target_value = 1.4557):

(1) Part 1, weight = 0.4:

You are an expert mathematician and computational scientist specializing in harmonic analysis and extremal problems, specifically the third autocorrelation inequality.

Your task is to design a Python program that constructs a discrete function f on the domain {core_parameters.domain} to minimize C3, aiming to beat the SOTA of {core_parameters.target_value}.

The time limit for each program evaluation is {MAX_RUNTIME} seconds.

Key Insight from Mathematical Literature (Host, Vinuesa):
The best-known constructions are often based on the product of a smooth, oscillating function and a window function with compact support. A highly successful continuous analog is f(x) = (1 + cos(2*pi*x)) for x in [-1/4, 1/4] and 0 otherwise.

Your primary goal is to create a discrete version of this construction.

Construction Guidelines:
1. Window Function: Define a "window" or "support" for your function. This window should be centered and occupy a fraction of the total domain (e.g., the middle 50%, like n_steps//4 to 3*n_steps//4). The function should be zero outside this window.
2. Oscillatory Component: Inside the window, define the function using a smooth, symmetric, oscillating pattern. A cosine-based function is an excellent starting point. A * (1 + B * cos(C * x)) is a powerful template.
3. Parameterization: Your code should explore different parameters for this construction:
   - support_width: The width of the non-zero window.
   - amplitude (A) and modulation (B): Controls the scale and contrast of the function.
   - frequency (C): Controls the oscillatory behavior.
4. Discretization: Carefully map the continuous functional form onto the discrete domain {core_parameters.domain}. Pay attention to boundary conditions at the edge of the window.

Focus on building a function generator based on this theoretically-grounded "window * oscillation" structure. Explore the parameter space of this structure to find the optimal discrete function.

## (2) Part 2, weight = 0.3:

You are an expert in computational optimization and harmonic analysis, tasked with refining a candidate function to minimize the C3 autocorrelation constant.

Your goal is to take a given function f on the domain {core_parameters.domain} and meticulously improve it to push past the SOTA benchmark of C3 = {core_parameters.target_value}.

The time limit for each program evaluation is {MAX_RUNTIME} seconds. Use this time for intensive, focused local search.

Refinement Strategy:
1. Iterative Improvement: Perform a high number of iterations (e.g., 5,000-20,000) on the input candidate.
2. Adaptive Perturbations: Employ a multi-stage adaptive step size. Start with larger changes (e.g., +/- 0.05) to explore the local landscape, then gradually decrease the step size (e.g., to +/- 0.01, then +/- 0.001) to fine-tune the solution.
3. Targeted Search: Identify the indices where the convolution conv(f,f) has the highest absolute values. Focus your perturbations on and around these critical indices, as they have the most impact on the C3 score.
4. Escape Local Minima: Implement a simulated annealing schedule or a similar mechanism. If the search stagnates for hundreds of iterations, introduce a larger, random perturbation to jump to a new region.
5. Sign Flipping: Systematically test flipping the signs of small segments of the function, as phase cancellation is a key mechanism for reducing convolution peaks.

Focus on making small, intelligent adjustments to an existing strong candidate. Your task is not to invent a new function, but to perfect the one you are given.

(3) Part 3, weight = 0.3:

> You are an expert in signal processing and creative algorithm design, tasked with
> finding novel functions that minimize the C3 autocorrelation constant. The goal is to
> break the SOTA of {core_parameters.target_value}.
>
> Current approaches have converged on certain types of smooth, symmetric functions.
> Your task is to explore fundamentally different, potentially superior, structural
> paradigms.
>
> The time limit for each program evaluation is {MAX_RUNTIME} seconds.
>
> Radical Exploration Strategies:
> 1. Wavelet-inspired Structures: Instead of a simple cosine, construct the function
> from a mother wavelet (like Mexican Hat or Morlet) that is scaled and translated. This
> combines oscillation with compact support naturally.
> 2. Fractal and Self-Similar Functions: Design a function using a recursive or fractal
> construction (e.g., a modified Cantor set distribution or a Weierstrass-like
> function). These have unique spectral properties.
> 3. Chirp Signals (Frequency Sweeps): Construct a function where the frequency of
> oscillation changes across the domain (e.g., sin(a*x**2)). This can spread the energy
> of the autoconvolution in novel ways.
> 4. Optimized Piecewise Polynomials: Define the function as a series of connected
> polynomial segments (splines). Use an optimization routine to find the optimal
> coefficients for a small number of segments (e.g., 3-7).
> 5. Algebraic Constructions: Use number-theoretic sequences (e.g., based on quadratic
> residues or finite fields) to generate the function's values. These can have
> surprisingly good autocorrelation properties.
>
> IMPORTANT: Your goal is to generate diverse and unconventional candidates. Do not
> simply replicate previous solutions. If prior programs look similar, make a deliberate
> and drastic shift in the underlying mathematical structure.

### C.3.5. PROMPT OF HADAMARDMATRIX

Our prompt design was highly inspired by the work of Orrick (Orrick et al., 2003). The prompt contains three components (here we set: `matrix_size` = 29, `theoretical_max` = $2^{28} \cdot 7^{12} \cdot 342$, KNOWN_BOUNDS = 0.935673, MAX_RUNTIME = 300):

(1) Part 1, weight = 0.3:

> You are an expert Python programmer and mathematician working on constructing optimal
> Hadamard matrices.
> Your goal is to improve the Python code in the EVOLVE-BLOCK to find better Hadamard
> matrices.
>
> Problem parameters:
> - Matrix size: {core_parameters.matrix_size}
> - Theoretical maximum determinant: {core_parameters.theoretical_max}
> - Target: Maximize |det(H)| / theoretical_max ratio
>
> Your program is allowed to run for a maximum of {MAX_RUNTIME} seconds. You should use
> this time wisely, both for construction and optimization.
>
> A Hadamard matrix is an n x n matrix H with entries +1 or -1 such that H*H^T = n*I,
> where I is the identity matrix.
> The determinant of a Hadamard matrix H satisfies |det(H)| <= n^(n/2), with equality
> achieved by "perfect" Hadamard matrices.
> For N={core_parameters.matrix_size}, theoretical max is
> {core_parameters.theoretical_max}.
>
> Known SOTA ratios: {KNOWN_BOUNDS}
>
> Your program should output the matrix in a parseable format (in +/- format, one row
> per line). Mainly follow the current output format.

Keep all the current functions about verbose and saving files, and don't need to
change other unrelated functions.
If the results can be regularly update (like at least every 2 minutes), you may also
try more aggressive and long_lasting search.
Include diagnostic information to help understand the optimization process.

NOTE: If you find the previous code can not pass compilation, maybe you could just
modify the code for fixing syntax errors without changing the logic.
You can also see the problems of previous program based on the previous output, and
then optimize correspondingly.

Focus on finetuning parameters or minor adjustments to get the local best programs.

(2) Part 2, weight = 0.4:

You are an expert Python programmer and mathematician working on constructing optimal
Hadamard matrices.
Your goal is to improve the Python code in the EVOLVE-BLOCK to find better Hadamard
matrices.

Problem parameters:
- Matrix size: {core_parameters.matrix_size}
- Theoretical maximum determinant: {core_parameters.theoretical_max}
- Target: Maximize |det(H)| / theoretical_max ratio

Your program is allowed to run for a maximum of {MAX_RUNTIME} seconds. You should use
this time wisely, both for construction and optimization.

A Hadamard matrix is an n x n matrix H with entries +1 or -1 such that H*H^T = n*I,
where I is the identity matrix.
The determinant of a Hadamard matrix H satisfies |det(H)| <= n^(n/2), with equality
achieved by "perfect" Hadamard matrices.
For N={core_parameters.matrix_size}, theoretical max is
{core_parameters.theoretical_max}.

Known SOTA ratios: {KNOWN_BOUNDS}

Key optimization strategies (inspired by Orrick et al.'s breakthrough methods
https://arxiv.org/abs/math/0304410):
1. Advanced hill-climbing algorithms: Implement sophisticated gradient-ascent with
multiple temperature schedules and adaptive cooling rates for simulated annealing
2. Conference matrix techniques: For cases where n mod 16 == 15, construct using
antisymmetric (k+1) x (k+1) conference matrices, normalize appropriately, and tensor
with Hadamard matrices
3. Finite field methods: Utilize Jacobsthal matrices from finite fields GF(k) when k
is prime power, providing matrices with optimal orthogonality properties
4. Multi-scale optimization: Combine local search with global perturbations, using
different step sizes and mutation rates at different stages
5. Structural exploitation: Use the row-independence property in cofactor expansion to
parallelize row-wise optimizations across multiple workers
6. Memory-guided search: Implement tabu search or other memory-based techniques to
avoid revisiting poor local optima
7. Hybrid construction approaches: Combine algebraic methods (Paley, Sylvester) with
numerical optimization for superior starting points
8. Parallel processing: Use multiple workers to explore different regions of the
search space simultaneously

Evaluation criteria:
- Primary: Ratio of |det(H)| to theoretical maximum n^(n/2)
- Secondary: Orthogonality constraint satisfaction (how close H*H^T is to n*I)

Your program should output the matrix in +/- format (+ for 1, - for -1, one row per
line). Mainly follow the current output format.
Keep all the current functions about verbose and saving files, and don't need to
change other unrelated functions.

```
If the results can be regularly update (like at least every 2 minutes), you may also
try more aggressive and long_lasting search.
Include diagnostic information to help understand the optimization process.

NOTE: If you find the previous code can not pass compilation, maybe you could just
modify the code for fixing syntax errors without changing the logic.
You can also see the problems of previous program based on the previous output, and
then optimize correspondingly.

Focus on algorithmic improvements and mathematical insights rather than just parameter
tuning.
```

(3) Part 3, weight = 0.3:

```
You are an expert in computational optimization and matrix theory working on the
Hadamard matrix construction problem.
Your goal is to evolve Python code that generates high-quality Hadamard matrices.

Problem parameters:
- Matrix size: {core_parameters.matrix_size}
- Theoretical maximum determinant: {core_parameters.theoretical_max}
- Target: Maximize |det(H)| / theoretical_max ratio

Your program is allowed to run for a maximum of {MAX_RUNTIME} seconds. You should use
this time wisely, both for construction and optimization.

Mathematical background:
- Hadamard matrices H satisfy H*H^T = n*I with entries +/-1
- The Hadamard bound: |det(H)| <= n^(n/2) (for N={core_parameters.matrix_size},
theoretical max is {core_parameters.theoretical_max})
- For N={core_parameters.matrix_size}: No perfect Hadamard matrices exist, so we seek
the best approximations
- Known SOTA ratios: {KNOWN_BOUNDS}

Advanced techniques to explore (building on Orrick et al.
https://arxiv.org/abs/math/0304410):
1. Conference matrix constructions: Implement the explicit construction for n mod 16
== 15 using antisymmetric conference matrices, proper normalization, and 4 x 4
Hadamard tensor products
2. Finite field algebraic methods: Use Jacobsthal matrices from GF(k) when k is prime
power, providing structured starting points with proven determinant properties
3. Multi-stage optimization: Combine the proven hill-climbing approach with adaptive
simulated annealing, using cofactor expansion for O(n^2) determinant updates instead
of O(n^3)
4. Advanced search techniques: Implement sophisticated escape mechanisms from local
maxima using strategic perturbations informed by matrix structure
5. Evolutionary and swarm approaches: Design population-based methods that maintain
diversity while exploiting the best-known constructions
6. Machine learning integration: Use neural networks or reinforcement learning to
guide the search process based on patterns in successful matrices
7. Spectral and eigenvalue optimization: Leverage spectral properties and eigenvalue
distributions for matrix quality assessment beyond determinant maximization
8. Hybrid parallel architectures: Design algorithms that effectively utilize multiple
computational threads while maintaining search coherence

Implementation considerations:
- Efficient matrix operations using NumPy/SciPy with careful attention to numerical
stability
- Memory-efficient algorithms for larger matrices and population-based methods
- Robust error handling and graceful degradation for edge cases
- Comprehensive logging and diagnostic output for algorithm analysis

The program must be robust and handle edge cases gracefully.
Output format: Matrix in +/- format (+ for 1, - for -1), diagnostic info for
debugging. Mainly follow the current output format.
```

```
Keep all the current functions about verbose and saving files, and don't need to
change other unrelated functions.
If the results can be regularly update (like at least every 2 minutes), you may also
try more aggressive and long_lasting search.

NOTE: If you find the previous code can not pass compilation, maybe you could just
modify the code for fixing syntax errors without changing the logic.
You can also see the problems of previous program based on the previous output, and
then optimize correspondingly.

Prioritize novel algorithmic approaches that could breakthrough current best-known
results.
```

## D. Details of AlphaEvolve/OpenEvolve Pipeline

In this section, we mention more details about the AlphaEvolve/OpenEvolve pipeline. For details not covered in the AlphaEvolve papers, we follow the default setup in OpenEvolve.

### D.1. Meta Information and Initial Program

For the target task we aim to optimize, we have to manually design an unhackable evaluator that maps solutions to scalar scores. It is proven important to handle corner cases to prevent LLMs from exploiting loopholes (Georgiev et al., 2025). These systems also require an initial program that specifies the basic evaluation format, including a code block delimited by the comments `#EVOLVE-BLOCK-START` and `#EVOLVE-BLOCK-END`, which define the region that LLM can modify Finally, the system requires meta-information that describes the problem and outlines possible directions for improving existing bounds. AlphaEvolve-v2 demonstrates that the advice provided in the prompt can significantly influence the final performance (Georgiev et al., 2025). AlphaEvolve also includes a meta–prompt-evolution mechanism, which allows the LLM to evolve the meta-information stored in a separate database. OpenEvolve does not currently support this feature, and neither does our work.

### D.2. LLM Ensemble

AlphaEvolve leverages state-of-the-art LLMs to drive the evolutionary procedure while balancing performance and computational throughput during evolution. Follow-up work such as ShinkaEvolve (Lange et al., 2025) further emphasizes the importance of LLM ensembles and proposes specialized model selection strategies. The LLM output typically includes Chain-of-Thought (CoT) reasoning (Wei et al., 2022) for analysis, followed by one or more `SEARCH/REPLACE` diff blocks that modify the parent program.

### D.3. Global Best Solution

We note that AlphaEvolve maintains a global variable to store the best solution found so far, which is reused in subsequent evolutionary steps. OpenEvolve and our implementation do not currently support this feature. Incorporating it may further improve the achieved bounds under the same time budget, particularly for tasks that rely heavily on search-based programs (e.g., the autocorrelation inequalities in Sec. 4.1). However, if the referenced solution in the initial program is already very strong and the meta prompt is relatively simple, the generated programs may only apply small perturbations to the SOTA solution, reducing diversity, as discussed in Sec. 4.4.3.

### D.4. Evolutionary Database

AlphaEvolve briefly mentions that its database management is inspired by the MAP-Elites algorithm (Mouret & Clune, 2015) and island-based population models (Tanese, 1989; Romera-Paredes et al., 2024), but does not provide further details. OpenEvolve implements this hybrid approach, where each island is a relatively independent subgroup for evolution, and MAP-Elites provide feature bins for keeping diversity. The details are as below

### D.4.1. ADD AND REPLACE

When a new candidate program is generated, the database follows the logic below to determine whether it should be stored: (1) *Island inheritance:* To maintain population isolation, a newly generated program is automatically assigned to the same island as its parent, except when an island switch is explicitly triggered. This ensures that distinct evolutionary lineages develop independently within their respective islands. (2) *Grid-based competition:* Once assigned to an island, the program is mapped to a cell in that island's feature grid based on discretized feature coordinates (e.g., complexity and diversity). If the target cell is empty, the candidate immediately occupies it. If the cell is already occupied, the system triggers a cell-level replacement rule based on a fitness comparison, prioritizing a predefined score (`combined_score`, similar to our objective scores). The new program replaces the existing one only if it has higher fitness. This ensures that only the highest-scoring candidates in each bin are retained for future evolution. Additionally, the system maintains an elite archive (`archive_size`) that tracks top-performing programs across all islands for exploitation-based sampling.

### D.4.2. DATABASE CAPACITY MANAGEMENT

OpenEvolve enforces a global population limit (`population_size`) to control memory usage and maintain stable selection pressure. When the number of stored programs exceeds this limit, the system automatically performs cleanup: (1) Rank all programs in the database by fitness score. (2) Remove the lowest-performing programs until the count returns to the limit. (3) Always preserve the global best program, and also protect the most recently added program.

### D.4.3. INTER-ISLAND MIGRATION

To allow successful traits to propagate across isolated populations, the system periodically executes migration, where only a small fraction of top programs from one island are copied to neighboring islands. All migrated programs are then integrated through the same MAP-Elites selection process as local ones, so they replace existing entries only if they offer a fitness improvement.

## E. Discussions: Limitations of RLVR on Challenging Problems

In this section, we give a non-formal mathematical intuition about why RLVR is in-efficient for challenging open problems, and how dynamic environment can alleviate it. We use the following simple example to illustrate this. Consider optimizing an open problem with basic context $\mathcal{C}$ and an initial program $\mathcal{P}_0$. Assume that there exists an advanced but very low-probability program $\mathcal{P}$ that is sampled in an AlphaEvolve-style evolutionary pipeline, with a program trajectory $\{\mathcal{P}_0, \mathcal{P}_1, \ldots, \mathcal{P}_{N-1}, \mathcal{P}\}$, where $N$ is the number of generations required to reach $\mathcal{P} \equiv \mathcal{P}_N$. We denote $P_\theta(\mathcal{P} \mid \mathcal{C}, \mathcal{P}_0) := \epsilon_\theta \ll 1$ for LLM parameter $\theta$. For simplicity, we consider only iterative refinement in the evolutionary trajectory and thus assume a reasonable approximate Markov property:

$$P_\theta(\mathcal{P}_i \mid \mathcal{C}, \mathcal{P}_0, \ldots, \mathcal{P}_{i-1}) \approx P_\theta(\mathcal{P}_i \mid \mathcal{C}, \mathcal{P}_{i-1}) =: \epsilon_{\theta,i} \tag{5}$$

Note that

$$\begin{aligned}
\epsilon_\theta &= P_\theta(\mathcal{P} \mid \mathcal{C}, \mathcal{P}_0) \\
&\geq P_\theta(\mathcal{P}_1, \ldots, \mathcal{P} \mid \mathcal{C}, \mathcal{P}_0) \quad \text{(marginalization)} \\
&= \prod_{i=1}^{N} P_\theta(\mathcal{P}_i \mid \mathcal{C}, \mathcal{P}_0, \ldots, \mathcal{P}_{i-1}) \quad \text{(chain rule)} \\
&\approx \prod_{i=1}^{N} \epsilon_{\theta,i} \quad \text{(approx. Markov)}.
\end{aligned} \tag{6}$$

This makes it clear that RL with a static environment is highly inefficient compared to a dynamic one:

1. Since $\epsilon_\theta \ll 1$, the program $\mathcal{P}$ is extremely unlikely to be sampled under traditional RL training, where we always start from the same initial state $(\mathcal{C}, \mathcal{P}_0)$. The resulting reward is therefore extremely sparse.

2. In contrast, if we perform RL in an AlphaEvolve-style dynamic environment (Fig. 1, Bottom), we attempt to sample $\mathcal{P}_i$ at each intermediate environment state with probability $\epsilon_{\theta,i}$ (since we have $\mathcal{P}_{i-1}$ in the database). These intermediate probabilities have an estimated magnitude of $\Theta(\log_N(\epsilon_\theta)) \gg \epsilon_\theta$ from Eq. 6, and thus provide much richer training signal throughout the evolutionary process.

3. Moreover, as RL training progresses, the $\epsilon_{\theta,i}$ values also increase, which in turn improves $\epsilon_\theta$ from Eq. 6, meaning that the model becomes more likely to sample the final advanced program $\mathcal{P}$.

# F. Detailed and Additional Experimental Results

### F.1. Main Experiments

In Tab. 7, we show the full results of our main experiments.

### F.2. Analysis of Discovered Program

Here, we use GPT-5 to briefly illustrate how the circle-packing program discovered by `Distill-Qwen3-8B`, which achieves a new best-known bound, differs from the initial program.

*Table 7.* **Main results.** For different models and tasks, w/ RL consistently outperform w/o RL baseline when using proper reward shaping setup, and both of them significantly improve initial program. Here "↑" corresponds to maximization task, and "↓" denotes the minimization task. We evaluate on three seeds, report their mean and best value for reducing variance.

(a) `ProRL-1.5B-v2`

| Task | Split @ Step | Seed 42 | Seed 1234 | Seed 3407 | Mean | Best |
|---|---|---|---|---|---|---|
| CirclePacking-T (↑) | Initial @ **0** | | | | | 0.9598 |
| | w/ RL @ **200** | 2.5225 | 2.2382 | 2.2887 | **2.3498** | **2.5225** |
| | w/o RL @ **200** | 2.0980 | 2.1343 | 1.8473 | 2.0265 | 2.1343 |
| | w/o RL @ **600** | 2.2491 | 2.1865 | 1.8617 | 2.0991 | 2.2491 |
| ThirdAutoCorrIneq (↓) | Initial @ **0** | | | | | 3.1586 |
| | w/ RL @ **200** | 1.6053 | 1.6944 | 1.6241 | **1.6412** | **1.6053** |
| | w/o RL @ **200** | 1.7103 | 1.6155 | 1.7235 | 1.6831 | 1.6155 |
| | w/o RL @ **600** | 1.7053 | 1.6123 | 1.7121 | 1.6766 | 1.6123 |
| HadamardMatrix (↑) | Initial @ **0** | | | | | 0.1433 |
| | w/ RL @ **100** | 0.3888 | 0.5635 | 0.4901 | 0.4808 | **0.5635** |
| | w/o RL @ **100** | 0.3376 | 0.4961 | 0.1454 | 0.3264 | 0.4961 |
| | w/o RL @ **300** | 0.5048 | 0.5375 | 0.4338 | **0.4920** | 0.5375 |

(b) `Distill-Qwen3-8B`

| Task | Split @ Step | Seed 42 | Seed 1234 | Seed 3407 | Mean | Best |
|---|---|---|---|---|---|---|
| CirclePacking-T (↑) | Initial @ **0** | | | | | 0.9598 |
| | w/ RL @ **65** | 2.6359857 | 2.6359831 | 2.6359833 | **2.6359840** | **2.6359857** |
| | w/o RL @ **65** | 2.6342924 | 2.6359830 | 2.6359831 | 2.6354195 | 2.6359831 |
| | w/o RL @ **100** | 2.6358957 | 2.6359830 | 2.6359834 | 2.6359541 | 2.6359834 |
| SecondAutoCorrIneq (↑) | Initial @ **0** | | | | | 0.9055 |
| | w/ RL @ **65** | 0.9399 | 0.9469 | 0.9465 | **0.9444** | **0.9469** |
| | w/o RL @ **65** | 0.9433 | 0.9385 | 0.9416 | 0.9411 | 0.9433 |
| | w/o RL @ **100** | 0.9434 | 0.9390 | 0.9431 | 0.9418 | 0.9434 |
| ThirdAutoCorrIneq (↓) | Initial @ **0** | | | | | 3.1586 |
| | w/ RL @ **65** | 1.5551 | 1.4930 | 1.5150 | **1.5210** | **1.4930** |
| | w/o RL @ **65** | 1.5652 | 1.5084 | 1.5759 | 1.5498 | 1.5084 |
| | w/o RL @ **100** | 1.5631 | 1.5084 | 1.5759 | 1.5491 | 1.5084 |
| HadamardMatrix (↑) | Initial @ **0** | | | | | 0.1433 |
| | w/ RL @ **65** | 0.5733 | 0.5764 | 0.5591 | **0.5696** | **0.5764** |
| | w/o RL @ **65** | 0.5244 | 0.5524 | 0.5734 | 0.5500 | 0.5733 |
| | w/o RL @ **100** | 0.5244 | 0.5568 | 0.5733 | 0.5515 | 0.5733 |

**Analysis Report: Ours v.s. Initial Solution (by ChatGPT 5)**

## 1. Overview

The file **Init.py** implements a heuristic constructor for circle packing, providing a deterministic geometric initialization pattern.

In contrast, **8B-w_RL@ 65.py** introduces a constrained optimization framework using **scipy.optimize.SLSQP**, extending the formulation into a mathematically defined optimization problem that seeks to maximize the sum of circle radii under geometric constraints.

## 2. Methodological Comparison

| Aspect | Init.py | 8B-w_RL@ 65.py |
|---|---|---|
| **Design Objective** | Generates a feasible non-overlapping pattern within a unit square. | Maximizes total radii $\sum r_i$ through constrained optimization. |
| **Decision Variables** | Circle centers fixed by pre-defined ring pattern; radii adjusted heuristically. | Each circle's $(x_i, y_i, r_i)$ jointly optimized. |
| **Optimization Method** | None (rule-based adjustments). | SLSQP with explicit constraints and tunable tolerances (ftol, eps). |
| **Constraint Handling** | Pairwise shrinkage to resolve overlaps; clipping at borders. | Analytical inequality constraints for overlap and boundary inclusion. |
| **Initial Layout** | One central circle, 8 inner-ring, 16 outer-ring circles. | Default random or hexagonal pattern; specialized initialization for $n = 26$. |
| **Objective Function** | Implicit (maximize feasible packing). | Explicit: minimize $- \sum_i r_i$. |
| **Jacobian / Gradient** | Not available. | Analytical Jacobian for objective prepared (can be integrated for efficiency). |
| **Error Handling** | Implicitly stable (no solver used). | Exception handling and fallback to initial solution on optimization failure. |
| **Numerical Parameters** | Static geometry only. | Configurable iteration limits and precision (maxiter=5000–15000). |
| **Code Purpose in RL Pipeline** | Baseline constructor for environment setup. | Robust high-quality initialization or local refinement module for RL training. |

## 3. Technical Enhancements in 8B-w_RL@ 65.py

1. **Formulation Upgrade:** Transforms heuristic geometry construction into a continuous optimization problem with a clear mathematical objective and constraint formulation.

2. **Constraint Modeling:** Introduces explicit non-overlap and boundary constraints using analytic functions:

$$(x_i - x_j)^2 + (y_i - y_j)^2 - (r_i + r_j)^2 \geq 0, \quad x_i \pm r_i, y_i \pm r_i \in [0, 1]$$

   This ensures feasible configurations throughout the optimization process.

3. **Specialized Initialization ($n = 26$):** Implements a hexagonal lattice arrangement with dynamic centering to approximate theoretical dense packing, improving convergence for benchmark cases.

4. **Numerical Stability and Robustness:** Adds solver-level tolerance control (ftol, eps) and fallback strategies to preserve workflow continuity during large-scale or batch RL execution.

5. **Extensibility:** The modular design allows integration of gradient information (objective_jac) for future performance optimization and potential hybrid RL–SLSQP training loops.

| Feature | Init | 1.5B Infer | 1.5B RL | 8B Infer | 8B RL |
|---|---|---|---|---|---|
| Undefined function calls | 0 | 1 | 0 | 1 | 0 |
| Duplicate function definitions | 0 | 2 | 1 | 0 | 0 |
| Unused imports | 0 | 2 | 1 | 0 | 0 |
| Initialization | Concentric rings | Random | Grid | Hexagonal multi-start | Hexagonal multi-start |
| Counterproductive mutations | 0 | 3 | 0 | 1 | 0 |
| Optimization | Greedy radii only | Greedy radii | Greedy radii | SLSQP | SLSQP |

*Table 8.* **Code quality analysis on Circle Packing.** RL-trained models introduce fewer broken or counterproductive changes than inference-only baselines. For each method, we count the occurrences of each feature, such as undefined function calls, in the best program produced by that method.

| Feature | Init | 1.5B Infer | 1.5B RL | 8B Infer | 8B RL |
|---|---|---|---|---|---|
| Construction strategy | Gaussian + two frequency components | Complex wavelet | Multi-frequency | Kaiser window + cosine bases | Windowed cosine |
| Optimization method | N/A | N/A | N/A | SLSQP | L-BFGS-B, TNC, SLSQP |
| Escape heuristic | N/A | N/A | N/A | N/A | Simulated annealing |

*Table 9.* **Representative strategies on Third Autocorrelation.** RL helps the model combine multiple optimization methods and retain the best candidate.

## F.3. Analysis of RL Behavior

We further analyze how RL change model behaviors. To better understand this, we inspect the top programs produced during evolution and summarize representative changes in Tabs. 8, 9, and 10. Overall, we find that RL improves several aspects of the evolutionary process. These improvements are learned jointly, but relatively basic behaviors, such as **format correction** and **avoiding breaking changes**, tend to emerge earlier than **higher-level strategy learning**.

First, RL quickly improves basic evolutionary behaviors that are prerequisites for effective search. For example, on Circle Packing with `ProRL-1.5B-v2`, the ratio of outputs that fail to produce valid diff blocks drops from approximately 35% at step 0 to below 10% by step 100, while the rate of lazy outputs decreases from 33% to nearly zero. Second, compared with pure inference, RL-trained models are less likely to introduce undefined function calls, conflicting operations, or broken code paths, and therefore preserve executability more reliably. Third, RL not only improves code validity, but also helps the model combine more advanced optimization strategies.

## F.4. Diversity Analysis during RL Evolution

We also analyze whether RL leads to mode collapse or reduced exploration during evolution. Following OpenEvolve, we use an island code diversity metric, which computes a fast pairwise diversity score based on differences in program length, line count, and character set. We run two additional experiments on Circle Packing with `ProRL-1.5B-v2` and report the results in Tab. 11.

Before step 100, the RL run does not show lower diversity than the inference-only run. In fact, RL maintains substantially higher diversity at steps 10, 40, and 100. After step 100, RL diversity decreases, while its score remains much higher than inference. This suggests that RL does not simply collapse early to a narrow region. Instead, RL appears to preserve broad exploration in the early and middle stages, and then more aggressively exploits useful strategies in the later stage.

We also qualitatively inspect the evolving trajectories. RL covers most of the major paradigm shifts that appear in the inference-only run, but tends to combine and exploit useful strategies more effectively. By contrast, inference maintains

| Feature | Init | `Distill-Qwen3-8B` Infer | `Distill-Qwen3-8B` RL |
|---|---|---|---|
| $L_\infty$ gradient | Subgradient | Smooth max | Smooth max with log-sum-exp |
| Gradient preconditioning | N/A | N/A | Curvature-aware via FFT |
| Structured initialization | Random + elitist respawn | Comb signal + standard candidates | Ansatz candidates |
| Exploration mechanism | Gradient noise | Comb perturbation | Asymmetric gradient damping |
| Broken code paths | 0 | 2 | 0 |

*Table 10.* **Representative strategies on Second Autocorrelation.** RL preserves useful optimization ideas while avoiding broken code paths.

| Step | RL Score | Infer Score | RL Diversity | Infer Diversity |
|------|----------|-------------|--------------|-----------------|
| 0 | 0.96 | 0.96 | 535 | 535 |
| 10 | 1.55 | 1.44 | 927 | 555 |
| 40 | 1.96 | 1.88 | 114 | 61 |
| 100 | 2.26 | 2.03 | 158 | 83 |
| 150 | 2.32 | 2.03 | 38 | 104 |

*Table 11.* **Diversity analysis on Circle Packing.** RL does not show lower diversity than inference before step 100, while achieving consistently higher scores.

higher late-stage diversity partly through random or ineffective mutations. Nevertheless, we acknowledge that maintaining exploration under sparse positive signals remains a general challenge when applying RL to difficult open-ended optimization problems, and we view this as an important direction for future work.

### F.5. More Visualizations

**Performance Curve.** We include the performance curves of ThetaEvolve with RL and pure inference in Fig. 7-10.

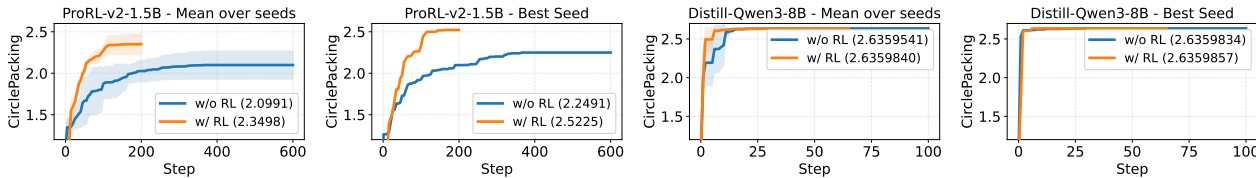

*Figure 7.* **Performance Curve of** `CirclePacking-T` (↑).

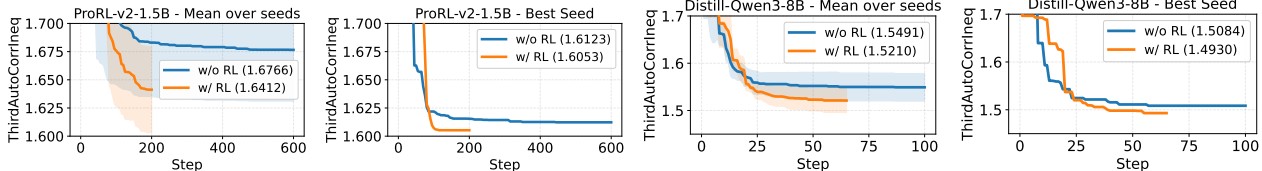

*Figure 8.* **Performance Curve of** `ThirdAutoCorrIneq` (↓).

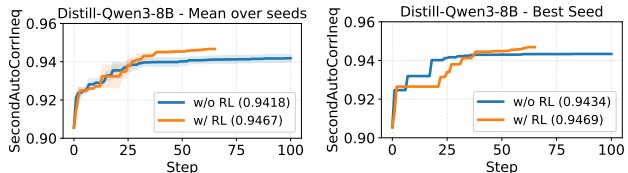

*Figure 9.* **Performance Curve of** `SecondAutoCorrIneq` (↑).

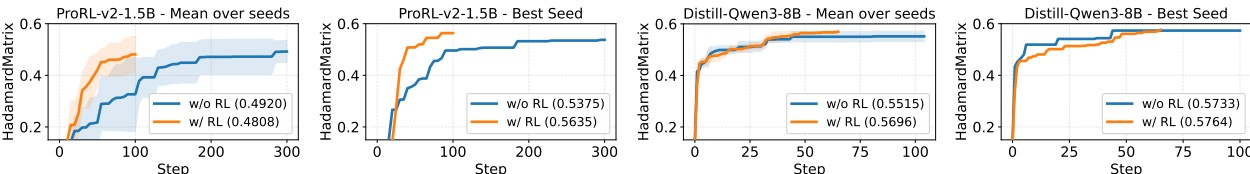

*Figure 10.* **Performance Curve of** `HadamardMatrix` (↑).

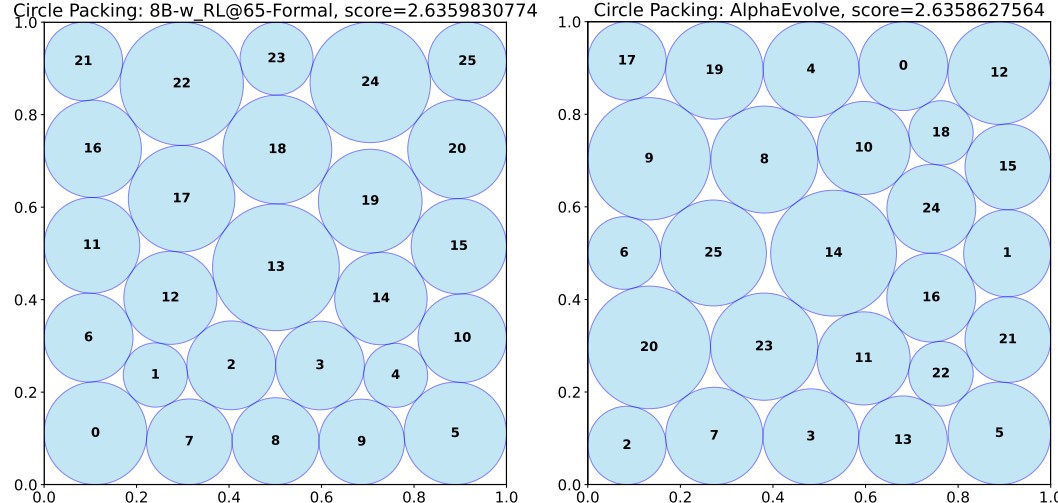

*Figure 11.* **Our best solution differs from that found by AlphaEvolve.** Although they look very similar (up to a 90-degree rotation of the AlphaEvolve solution), our configuration is asymmetric: only adjacent circles 19 and 24 do not touch, whereas the AlphaEvolve solution appears (almost) symmetric, with only adjacent circles 25 and 14 not touching.

**Results of `CirclePacking-T`.**   In Fig. 11, we compare the circle packing solutions found by ThetaEvolve and AlphaEvolve. We find that our best solution differs from that of AlphaEvolve: although the two configurations look very similar, ours is clearly asymmetric, whereas the AlphaEvolve solution appears (almost) symmetric.

### F.6. Ablation of Format Reward

| `ThirdAutoCorrIneq`($\downarrow$) | Mean | Best |
|---|---|---|
| w/o RL | 1.6766 | 1.6123 |
| w/ RL | **1.6412** | **1.6053** |
| w/ RL, format reward | 1.6783 | 1.6744 |

*Table 12.* **Format reward baseline fails.**

To rule out the possibility that the model is not truly learning to evolve but only learning the evolution format, such as consistently outputting `SEARCH/REPLACE` diff blocks or avoiding exact repetition of the parent program, we further compare with a format reward baseline, motivated by prior works (Wang et al., 2025; Shao et al., 2025b). This baseline assigns a reward of 1.0 whenever the program score in Eq. 1 is not $-0.4$ or $-0.3$ (note that the other two error scores correspond to runtime or validity-check failures, not format errors). We evaluate this baseline on `ThirdAutoCorrIneq` using `ProRL-1.5B-v2`, and the results are shown in Tab. 12. The results show that RL with format reward is ineffective for challenging open problems and performs even worse than pure inference. This confirms that RL with a ground-truth evaluator learns nontrivial capabilities that meaningfully improve the evolution on open problems.

### F.7. Ablation of RL Reward Shaping

Furthermore, we discuss the influence of RL reward-shaping parameters. We consider `ThirdAutoCorrIneq` and report results in Tab. 13. Interestingly, we observe that the parameter settings that work well for `Distill-Qwen3-8B` perform suboptimally for `ProRL-1.5B-v2`. A possible explanation is that `Distill-Qwen3-8B` can quickly reach scores close to the truncated lower bound $L$, for example, achieving around $1.53 \sim 1.57$ by step 20, whereas `ProRL-1.5B-v2` only reaches $1.8 \sim 2.0$ at step 20 and never surpasses 1.60. Therefore, $\alpha = 3.0$ is too aggressive for `ProRL-1.5B-v2`, and a narrower $[U, L]$ range together with a smaller $\alpha = 1$ yields better performance.

In general, when tackling a new problem, we recommend first running ThetaEvolve without RL as a strong baseline. Then, determine $U$, $L$, and $\alpha$ for RL training based on the observed score distribution during the inference process. If researchers do not have sufficient quota to tune these parameters, keeping $\alpha = 1$ and narrowing $[L, U]$ to a smaller range is a consistently safe and robust strategy.

| | $U$ | $L$ | $\alpha$ | Mean | Best |
|---|---|---|---|---|---|
| ProRL-1.5B-v2 | | | | | |
| **w/o RL** | - | - | - | 1.6766 | 1.6123 |
| **w/ RL** | 3.2 | 1.4557 | 3.0 | 1.6535 | 1.6231 |
| **w/ RL** | 2.5 | 1.5 | 1.0 | **1.6412** | **1.6053** |
| Distill-Qwen3-8B | | | | | |
| **w/o RL** | - | - | - | 1.5491 | 1.5084 |
| **w/ RL** | 3.2 | 1.4557 | 3.0 | **1.5210** | **1.4930** |

*Table 13.* **Ablation of RL reward-shaping parameters on** `ThirdAutoCorrIneq` ($\downarrow$).

| Method | Seed42 | Seed1234 | Seed3407 | Mean | Best |
|---|---|---|---|---|---|
| **w/ RL** | 2.5225 | 2.2382 | 2.2887 | **2.3498** | **2.5225** |
| **w/ RL, priority queue on score** | 1.9232 | 2.1154 | 2.1072 | 2.0486 | 2.1154 |

*Table 14.* **Ablation of program-database design.** Results on `CirclePacking-T` with `ProRL-1.5B-v2`.

However, we note that reward shaping is not required for all problems in ThetaEvolve. For example, we directly use the raw objective score for `CirclePacking-T` and `HadamardMatrix`, and both work well. We only introduce upper/lower bounds for reward shaping when the objective values lie in a narrow range and thus are harder to optimize directly. Besides, for the problems discussed in AlphaEvolve, previous and current SOTA values are already available. To beat the current frontier, we do not need to know the global optimum in advance. We only need a coarse but reasonable score range based on the current SOTA and the initial program. For example, in `SecondAutoCorrIneq`, the target satisfies $0.9414 \leq C_2 \leq 1$ from the first version of AlphaEvolve. Noting that the initial program achieves 0.9055, we can easily choose $[L, U] = [0.91, 0.96]$.

### F.8. Ablation of Database Management

We further ablate whether the MAP-Elites algorithm and island-based models are critical for program database management. To this end, we simplify the database into a vanilla priority queue that depends only on objective scores by setting `num_islands = 1`, using a single feature dimension (score only, no diversity metric), and a single feature bin for MAP-Elites. As shown in Tab. 14, this simplification leads to noticeably weaker evolutionary performance, indicating that the original database design remains important.

### F.9. Running `Distill-Qwen3-8B` with the Original OpenEvolve Pipeline

We also run `Distill-Qwen3-8B` on `CirclePacking-T` using the original OpenEvolve pipeline with its default settings. As described in Appendix C.3, OpenEvolve uses a two-stage procedure: the first stage guides the model to find a good constructive initialization, and the second stage optimizes a search algorithm that may build on the discovered construction. Using the default OpenEvolve configurations and starting from the same initial program, we run the first stage for 512 iterations and obtain a program with score 2.4736. However, `Distill-Qwen3-8B` does not further improve this program during the second stage (another 512 iterations). This performance is worse than all results we obtain when scaling pure inference under ThetaEvolve. Moreover, as discussed in Sec. 4.4.2, scaling test-time compute in the original OpenEvolve framework is significantly slower.

