# OpenReview forum: "ThetaEvolve: Test-time Learning on Open Problems"
_ICML.cc/2026/Conference — ICML 2026 regular_

### Official Review · Reviewer_nhyT · 2026-02-24

**Soundness:** 3
**Presentation:** 2
**Significance:** 3
**Originality:** 2
**Overall Recommendation:** 4
**Confidence:** 4

**Summary:**

The authors identify two major limitations in current evolutionary coding agents for open problems. First, most existing methods are either closed-source or overly complex. Second, current SOTA systems rely on ensembles of proprietary models, leaving trainable small models and inference-time parameter updates underexplored.

This work introduces ThetaEvolve. By integrating test-time RL training with evolutionary program generation, ThetaEvolve boosts exploration on open problems. Using a small model, this method outperforms previous SOTA results on circle packing and the first autocorrelation inequality while remaining significantly more efficient in wall-clock time.

**Compliance With Llm Reviewing Policy:**

Affirmed.

**Final Justification:**

My concerns have been adequately addressed.

**Key Questions For Authors:**

What is the motivation behind using raw objective values rather than their improvement (children over parents) as a reward? Have you tried the latter?

**Limitations:**

The authors have discussed the limitations of RLVR on challenging problems. Yet, the limitations of ThetaEvolve have not been adequately addressed.

**Strengths And Weaknesses:**

## Strengths

It introduces the first open-source framework that enables small models like DeepSeek-R1-Distill-Qwen-8B to surpass human-best results and match frontier models on open optimization problems.

The system successfully transitions from pure inference to test-time reinforcement learning, allowing models to internalize and generalize evolving strategies across different mathematical tasks.

Practical engineering innovations, including batch sampling, lazy penalties, and a large program database, significantly improve computational throughput and exploration efficiency compared to previous closed-source systems.

## Weaknesses

The methodology is not sufficiently clear. Specifically, it remains ambiguous how ThetaEvolve orchestrates the interplay between the evolutionary process and model training.

Prior research has already explored the co-evolution of model parameters and programs. It would be beneficial to include a comparative discussion with these works (e.g., [1]) in the main body of the paper to better highlight the unique contributions of this framework.

The experiments rely on relatively small models (1.5B and 8B), which often struggle to output valid diff blocks or executable code without specialized fine-tuning. This raises the possibility that the observed performance gains stem primarily from the model learning basic syntax and formatting rather than complex reasoning for evolution. An in-depth analysis is needed to determine what specifically constitutes the learned "evolutionary capabilities."

The framework requires large-scale program generation and evaluation (hundreds of thousands of programs). While this is feasible for open problems with low evaluation costs, it may be impractical for domains where evaluation is expensive (e.g., training a neural network as part of the fitness function). Some related works prioritize sample efficiency, limiting evaluations to ~100 programs [2]. It remains unexplored whether this framework remains effective under a more constrained computational budget.

[1] Huang, Z., Wu, W., Wu, K., Wang, J., & Lee, W. B. (2025). Calm: Co-evolution of algorithms and language model for automatic heuristic design. arXiv preprint arXiv:2505.12285.

[2] Ye, H., Wang, J., Cao, Z., Berto, F., Hua, C., Kim, H., ... & Song, G. (2024). Reevo: Large language models as hyper-heuristics with reflective evolution. Advances in neural information processing systems, 37, 43571-43608.

---

> ### Author Rebuttal · Authors · 2026-03-31
>
> Thanks a lot for your recognition and your valuable comments and suggestions. We respond to your questions below and would appreciate it if you could let us know if our response addresses your concerns.
>
> > **Q1**: how ThetaEvolve orchestrates the interplay between evolutionary process and model training.
>
> **A1**:
> We will clarify this in the revision:  Evolution and RL share the same data pipeline. In each step, B parent programs are sampled from the database and be built as prompt, and LLM generates N responses per prompt. The resulting child programs are evaluated and inserted back into the database. The evolutionary process happens in the database, which re-organizes programs using AlphaEvolve-style mechanisms such as population algorithms to retain high-performing and diverse candidates. RL is an optional optimization on model weights using the same prompt-response pairs. Unlike standard RL, the prompt distribution in ThetaEvolve is changing with evolution.
>
> > **Q2**:include a comparative discussion(e.g., [1]) to better highlight the unique contributions of this framework.
>
> **A2**: Thanks a lot for pointing this out! We will include a discussion of CALM [1] in the revision. CALM also co-evolves model parameters and programs, but there are several important differences.  (1) CALM mainly studies automatic heuristic design for standard combinatorial optimization benchmarks with training/test sets. In contrast, ThetaEvolve targets improving SOTA on open optimization problems without training sets, resulting in test-time learning on a single open problem. ThetaEvolve also supports AlphaEvolve-style full-program modification, resulting in a broader search space. (2) ThetaEvolve emphasizes scaling test-time compute, including larger program databases and batch sampling.
> (3) ThetaEvolve introduces mechanisms such as lazy penalties, which are important when the search is already close to the current SOTA.
>
> > **Q3**: The experiments rely on small models , which often struggle to output valid diff blocks …. analysis what constitutes "evolutionary capabilities."
>
> **A3**:
> Thanks for the constructive suggestion.Appendix F.4 (Table 8) includes an ablation where RL uses only a **format reward**, meaning the model gets a score of 1 only if it outputs a correct diff block and a child program different from the parent. This performs much worse than RL with outcome-based reward, showing that the model learns signals beyond formatting.
>
> We also conduct qualitative analysis. Due to space limits, please refer to Tables R1–R3 in our response to Reviewer L9JM. In short, the model learns both from errors and from better strategies during evolution, including format correction, fewer breaking changes, and stronger strategy composition. We believe larger models also share these behaviors.
>
> > **Q4**: …impractical when evaluation is expensive (e.g. training NN). Some works prioritize sample efficiency…[2]. whether it remains effective under a constrained budget.
>
> **A4**: We agree that evaluation cost and sample efficiency are important concerns, and related work like ReEvo[2] shows that techniques like reflection can improve sample efficiency.
>
> However, our focus is on a different and largely orthogonal dimension. Rather than optimizing for a low-budget regime, ThetaEvolve studies how program evolution improves as test-time compute scales, and how this can be combined with RL.  We show that **compute scaling itself is an important factor for evolving on open problems**. ThetaEvolve also does not change the underlying evolving algorithm in OpenEvolve, so techniques that improve sample efficiency are in principle compatible with our framework.
>
> More broadly, the right design depends on the bottleneck. For open math problems with cheap verification, more sampling with a smaller model may be effective. For settings with expensive evaluations, one should likely reduce evaluations and invest more in stronger models or more sophisticated search. This tradeoff is an important future direction.
>
> > **Q5**: using improvement (children over parents) as a reward?
>
> **A5**: Good question. We did consider using improvement-based rewards but we found some problems.
> (1) Child-minus-parent improvement creates a scale mismatch with the fixed error penalties in Eq. 1. On Circle Packing, an improvement can be 0.5 at an early stage (1.0 to 1.5), but only 0.01 later (2.30 to 2.31), even though the latter may be harder to obtain. With fixed penalties such as -0.2 for runtime errors, rewards like 0.01 may make the model overly conservative at later stages.
> (2) We also tried a binary improvement reward, assigning 1.0 whenever the child is better than the parent. But this performed slightly worse than raw objective. On Circle Packing with 1.5B, it achieved a mean score of 2.22, compared with 2.35 for RL with raw objective in Table 2.
> We agree that more advanced reward designs are possible, but view this as a separate future direction.

---

> > ### Author Rebuttal · Reviewer_nhyT · 2026-04-03
> >
> > My concerns have been adequately addressed. One quick follow-up: the prompt distribution in ThetaEvolve is changing with evolution. Would this cause any problems beyond conventional LLM RL?

---

> > > ### Author Response · Authors · 2026-04-04
> > >
> > > Thanks for your recognition and we are glad that our response helped address your concerns! The follow-up question about the changing prompt distribution is insightful and we detail our response as below.
> > >
> > > As discussed in the paper and our rebuttal, the changing prompt distribution is exactly what allows the model to keep learning at its current capability frontier and achieve much stronger performance. On the other hand, it can also introduce additional challenges.
> > >
> > > One direct effect is that the variance across runs can become larger for a static input distribution, since the later input distribution depends on earlier rollouts. In other words, randomness in the early stage can affect which regions of the search space the model sees later. This is one reason why we report multiple runs in the paper rather than relying on a single trajectory.
> > >
> > > Another difference from conventional RL is that, in standard LLM RL, we often care about the final best checkpoint itself. In ThetaEvolve, however, because the original model can already write basic programs reasonably well, the model at very late training stages may over-specialize to the final tuning strategies. As a result, the best checkpoint may appear in the middle or mid-last of training rather than at the very end, which needs some additional evaluation for selection. One possible way to mitigate this is to mix older programs from the database, similar to a replay-buffer effect, so that training is not dominated only by the latest frontier.
> > >
> > > Nevertheless, for open problems, the primary goal is still to achieve new SOTA rather than to obtain a single final checkpoint. Empirically, RL with the evolving system performs much better than conventional static-environment LLM RL on these tasks. We believe how to improve these RL in dynamic environments would be very interesting future directions.

---

### Official Review · Reviewer_S2vn · 2026-03-10

**Soundness:** 3
**Presentation:** 3
**Significance:** 2
**Originality:** 3
**Overall Recommendation:** 3
**Confidence:** 4

**Summary:**

This paper introduces ThetaEvolve, a framework designed to enhance AlphaEvolve by scaling test-time compute. The core mechanisms involve batch sampling for engineering acceleration, alongside the integration of test-time Reinforcement Learning and reward shaping.

**Compliance With Llm Reviewing Policy:**

Affirmed.

**Key Questions For Authors:**

Reference weaknesses.

**Limitations:**

The paper lacks a thorough discussion of limitations. Specifically, the authors should analyze the trade-off between the computational overhead of LLM parameter updates and the resulting marginal performance gains.

**Strengths And Weaknesses:**

Strengths

+ By parallelizing the heuristic evolutionary search via system-level batch sampling, the authors achieved a significant speedup, reducing runtime from 63.6 hours to just 5.4 hours (as shown in Table 5).
+ The study provides empirical evidence that scaling the program database with a single Large Language Model (LLM) effectively enhances performance in mathematical discovery tasks.
+ The cross-task generalization observed in Section 4.3.1 is noteworthy, demonstrating that test-time RL updates not only boost target task performance but also accelerate convergence on previously unseen tasks.

Weaknesses

+ The algorithmic novelty is limited. As Section 4.4.2 reveals, baselines like OpenEvolve achieve comparable performance gains when provided with an equivalent compute budget. This suggests that the observed improvements stem primarily from engineering-driven scale-up (via batch sampling) rather than inherent advances in the evolutionary mechanism.
+ While test-time RL yields additional gains, it introduces the overhead of model updates. The authors fail to demonstrate whether the marginal benefits of this complex RL optimization outweigh simply allocating the same wall-clock time or FLOPs to a larger-scale heuristic search.
+ The reward shaping strategy relies on manually predefined upper and lower bounds for each task. This approach is impractical for real-world open-ended optimization or mathematical discovery, where the global optimum is typically unknown  in advance.
+ The evaluation lacks a head-to-head comparison with current State-of-the-Art methods in LLM-based heuristic search and mathematical discovery, particularly under a normalized computational budget.

---

> ### Author Rebuttal · Authors · 2026-03-31
>
> Thank you for your recognition and your feedback. We have responded to your concerns and will revise our paper accordingly. We would appreciate it if you could let us know if our response addresses your concerns.
>
> > **Q1**: algorithmic novelty is limited. … This suggests improvements stem primarily from engineering-driven scale-up rather than inherent advances in evolutionary mechanism.
>
> **A1**:
> We respectfully disagree with the concern. We claim that **novelty should not be judged only by changing the evolutionary algorithm itself**.
> Sec. 4.4.2 shows that when given sufficient compute, even OpenEvolve can achieve stronger results (2.18) than its default low-budget setting (1.26). We view this not as weakening our contribution, but as supporting one of our main claims: for program evolution on open problems, scaling test-time computes is a first-order factor and has been underexplored in prior works. ThetaEvolve contributes a practical scaling recipe that makes such scaling feasible, substantially improving throughput while achieving new bounds. Under the same budget, OpenEvolve still underperforms ThetaEvolve inference (2.18 vs. 2.25) and requires much longer wall-clock time (63.6 vs. 5.4h), while ThetaEvolve with RL further improves to 2.52.
> ThetaEvolve also enables test-time RL in a dynamic evolution environment, so that model can learn to operate at the frontier of its evolving process and even generalize to other tasks. This aspect is not supported by OpenEvolve.
> Moreover, our work keeps the OpenEvolve evolutionary mechanism intact, showing that our framework is compatible with future advances in evolving algorithms.
>
> > **Q2/Limitation**: While test-time RL yields gains, it introduces overhead of model updates. The authors fail to shows whether marginal benefits of complex RL outweigh simply allocating the same wall-clock time or FLOPs to a larger-scale heuristic search.
>
> **A2**:
> First, for challenging open problems, achieving new SOTA always has the highest priority. Even the minor  frontier gains can be very valuable. In AlphaEvolve, a minor improvement on allocation algorithms can yield large cost savings for data centers, and that in bounds of kissing number can be a breakthrough in decades.
>
> Besides, RL already shows higher sample efficiency. Like on CirclePacking with 1.5B, the RL curve at around 100 steps already surpasses inference runs at 600 steps, and clearly from Figure 5, inference runs already goes to some plateau. RL consistently outperforms inference-only baselines across tasks with fewer steps, and RL would let models learn better behaviors.  On the other hand, wall-clock time or FLOPs are not very precise metrics for open optimization problems. Similar to us, AlphaEvolve/OpenEvolve also report on the evolution steps as a progress metric. Because as evolution progresses, stronger candidate programs often become more expensive to evaluate (e.g., simple initialization takes < 1s to eval while complex optimization takes longer like >60s), and their corresponding model outputs also tend to be more complex. So our measure is reasonable.
>
>
> > **Q3**: reward shaping relies on predefined bounds. This approach is impractical for real-world open-ended optimization… where global optimum is unknown in advance.
>
>
> **A3**:
> We kindly disagree that it is impractical. **For open optimization problems, we do not need global optimum to set this range, but the current SOTA we want to beat is enough**.
> This applies to all the open problems in AlphaEvolve,  and they always have some theoretical upper/lower bounds. Usually, a reasonable score range can be decided from prior literature and the initial program (Appendix B).
> For example, in SecondAutoCorrIneq, the target satisfies $0.9414 \le C_2 \le 1$ from the first version of AlphaEvolve. Noting that the initial program achieves 0.9055, we can easily choose $[L,U]=[0.91,0.96]$ for $0.96 \in [0.9414, 1]$.
>
>
> > **Q4**: The evaluation lacks a head-to-head comparison with current SOTA methods in LLM-based heuristic search and mathematical discovery, particularly under a normalized computational budget
>
> **A4**:
>
> To the best of our knowledge, ThetaEvolve is the first work that combines an AlphaEvolve/OpenEvolve-style evolving system with test-time RL in a dynamic evolution environment. Since ThetaEvolve achieves new best-known bounds on open problems studied in AlphaEvolve, we believe these comparisons are sufficient to support our main claims, and cover the necessary baselines established before ICML submission timeline.
>
> Moreover, our focus is to emphasize the importance of scaling test-time compute and enabling RL training on evolving systems, rather than optimizing purely for sample efficiency under a tightly budget. Although it is important to design better evolving algorithms to improve sample efficiency, it is largely orthogonal to our pipeline (we do not change the evolving algorithms from AlphaEvolve) but is outside the scope of our current focus.

---

> > ### Author Rebuttal · Reviewer_S2vn · 2026-04-01
> >
> > I appreciate the authors' clarifications and have increased my scores for Soundness and Originality to reflect the validated engineering contributions and RL efficiency. However, these merits do not fully offset my primary concerns regarding the lack of normalized budget comparisons (Q4) and the reliance on predefined reward bounds (Q3). Therefore, I maintain my overall recommendation of Weak Reject.

---

> > > ### Author Response · Authors · 2026-04-02
> > >
> > > Thanks very much for your appreciation and quick reply. We are glad that our rebuttal helped clarify the soundness and originality of the work. Below, we further address the remaining concerns about comparisons and predefined bounds.
> > >
> > > > **Q4-2**: Lack of budget comparison with current State-of-the-Art methods
> > >
> > > **A4-2**: As mentioned in our rebuttal, ThetaEvolve achieves new SOTA on open problems studied in AlphaEvolve. To the best of our knowledge, there is no concurrent SOTA work as defined by ICML2026 rules that we need to compare. We would sincerely appreciate it if you believe there is such a method and could point us to the specific current SOTA work.
> > >
> > >
> > > We also want to emphasize that improving sample efficiency through better evolving algorithms is largely orthogonal to our contributions. We do not change the MAP-Elite or population-based evolving algorithms from AlphaEvolve/OpenEvolve, so improved evolving algorithms can directly replace them and remain compatible with our framework.
> > >
> > > > **Q3**: Predefined task-specific reward bounds are impractical for real-world open-ended optimization or mathematical discovery, where the global optimum is unknown in advance
> > >
> > > **A3**:
> > > First, **reward shaping is not required for all problems in ThetaEvolve**. For example, we directly use the raw objective score for Circle Packing and Hadamard Matrix, and both work well. We only introduce upper/lower bounds for reward shaping when the objective values lie in a narrow range and thus are harder to optimize directly.
> > >
> > > Second, **for the problems discussed in AlphaEvolve, previous and current SOTA values are already available. To beat the current frontier, we do not need to know the global optimum in advance**. We only need a coarse but reasonable score range based on the current SOTA and the initial program. We have mentioned the example of second autocorrelation inequality in the previous response.
> > >
> > > To further support that it is a general case, we construct initial programs for several other AlphaEvolve problems and show that setting such ranges is straightforward:
> > >
> > > (1) **Uncertainty inequality**. This problem minimizes an upper bound on the uncertainty constant \(C_4\). The known theoretical range is \(0.2025 \leq C_4 \leq 0.3523\), where 0.3523 is the pre-AlphaEvolve SOTA, and AlphaEvolve improves it to 0.3216. Our initial program with vanilla Hermite coefficients obtains 0.3945. Therefore, a simple range such as \([L, U] = [0.31, 0.40]\) is already reasonable.
> > >
> > > (2) **Erdős minimum overlap problem**. This problem minimizes an upper bound on the constant \(C_5\). The known theoretical range is \(0.379005 \leq C_5 \leq 0.380927\), and AlphaEvolve improves the upper bound slightly to \(0.380924\). Our initial program obtains 0.3816. Hence, we can define a valid range such as \([L, U] = [0.3809, 0.3816]\), and narrow it further once a better program is found.
> > >
> > > (3) **Sums and differences of finite sets**. This problem maximizes a lower bound on the exponent \(C_6\). The previous lower bound is 1.14465, and AlphaEvolve improves it to 1.1584. Our initial program obtains 1.127. Thus, a simple range such as \([L, U] = [1.12, 1.17]\) is easy to choose.
> > >
> > > (4) **Packing unit regular hexagons inside a regular hexagon**. This problem minimizes the side length of the outer hexagon. For the \(n=11\) instance, the previous SOTA is 3.943, and AlphaEvolve improves it to about 3.931. Our initial program obtains 4.248. Therefore, a range such as \([L, U] = [3.90, 4.24]\) is again straightforward.
> > >
> > > Importantly, **these ranges do not require knowledge of the global optimum**. They only need to be reasonable enough to distinguish progress near the current frontier by not staying too far away from the current SOTA. If a run already exceeds the chosen bound, that itself indicates a new SOTA or near-SOTA result, and the range can be tightened in a later run if we gets better programs later and wants to use it as a new initial program.
> > >
> > > Finally, we believe **some task-specific design is often unavoidable in challenging open-ended optimization**. For example, recent AlphaEvolve-style work on Ramsey number bounds uses a hand-designed heuristic scoring function that combines graph size and conflict ratios to evaluate progress[1]. This is also a task-specific design choice. In this sense, setting a coarse reward range is a lightweight technique of the broader task-specific designs commonly needed in open problem discovery.
> > >
> > > We hope this helps address your concerns, and we will add these discussions to the revised paper. Please let us know if you have further questions.
> > >
> > >
> > > [1] Nagda, Ansh, Prabhakar Raghavan, and Abhradeep Thakurta. "Reinforced Generation of Combinatorial Structures: Ramsey Numbers." arXiv preprint arXiv:2603.09172 (2026).

---

### Official Review · Reviewer_L9JM · 2026-03-13

**Soundness:** 3
**Presentation:** 2
**Significance:** 3
**Originality:** 3
**Overall Recommendation:** 4
**Confidence:** 2

**Summary:**

This work proposes a new open-source framework, called thetaEvolve, that incorporates iterative refinement during the evolution process on open mathematical optimization problems. Overall, ThetaEvolve reduces complexity by using a single model, employs a larger database, and uses RL to distill good populations. With GRPO, lazy penalties are introduced to discourage the model from outputting redundant or non-improving code. Experiments show his test-time RL framework can match or surpass frontier multi-model ensembles, achieving competitive results on challenging tasks like Circle Packing.

**Compliance With Llm Reviewing Policy:**

Affirmed.

**Final Justification:**

The rebuttal addresses my concerns. I will keep my score at weak accept. I am positive overall, but I still see some room to improve the clarity of the practical discussion in the paper

**Key Questions For Authors:**

- Did the authors observe any form of mode collapse or reduced exploration under RL? It would be helpful to verify this with a diversity-related metric over training.

**Limitations:**

yes

**Strengths And Weaknesses:**

### Strengths

- The authors provide a highly valid and well-motivated problem formulation. Beyond static inference-only search, integrating RL directly into the evolving process is an interesting and practical direction.
- Strong empirical results with useful systems contribution: The reported performance is impressive, especially given the use of a relatively small open model. With a massive program database and batch sampling, the paper provides an efficient and practical framework for scaling test-time compute

### Weaknesses

- The paper argues that RL improves the model beyond inference-only search, but it is not fully clear how much of the gain comes from RL itself versus the larger program database.
- Figure 3 suggests that the benefit of RL is not uniform over the search horizon: under limited budgets, inference-only search can be competitive or even better in the early stage, while RL mainly improves later-stage or final performance. The manuscript would benefit from discussing this behavior more explicitly.
- While the authors use a large database to mitigate this issue, the manuscript lacks a rigorous empirical analysis of how solution diversity evolves during training. Providing a diversity-related metric over time would help confirm that the model is truly exploring the search space.

---

> ### Author Rebuttal · Authors · 2026-03-31
>
> Thank you for your recognition of our work and your constructive feedback to improve our paper. We will revise our paper based on your advice. We detail our response below and please kindly let us know if our response addresses your concerns.
>
>
> > **Q1**: not clear how much of the gain comes from RL versus larger database.
>
> **A1**:
> We disentangle the effects in separate experiments.
>
> (1) In Fig. 5, we show that simply scaling the program database improves performance. E.g., On Circle Packing with 1.5B, increasing database size yields 0.4 average improvement.
>
> (2) Table 2 shows that under the same program database setup, RL consistently outperforms pure inference. On a setup similar to (1), RL improves the average score by 0.25 with 1/3 steps.
>
> Since larger database improves diversity and exploration, while RL helps the model internalize strategies that reduce errors and improve search, their improvements are relatively orthogonal.
>
>
> > **Q2**: …benefit of RL is not uniform over the search horizon…discussing RL behavior more explicitly.
>
>
> **A2**:
> Thank you for this important point. To better analysis, we examined both the top programs during evolution. The main findings are summarized below, with examples in Tables R1–R3. In general these behaviors are learned at the same time, but relatively (1)(2) happens earlier than (3).
>
> (1) **Format correction.** RL quickly improves basic evolutionary behaviors that are prerequisites for effective search. For example, on Circle Packing with 1.5B, the ratio of outputs that fail to produce valid diff blocks drops from ~35% at step 0 to <10% by step 100, and the rate of lazy outputs decreases from 33% to near zero.
>
> (2) **Fewer breaking changes.** Compared with pure inference, RL-trained models are less likely to introduce undefined function calls, conflicting operations, or broken code paths, and thus preserve executability more reliably.
>
> (3) **Stronger strategy learning.** RL not only improves code validity, but also helps the model combine more advanced techniques.
>
>
> **Table R1**: Code quality on Circle Packing.
> |Feature|Init|1.5B Infer|1.5B RL|8B Infer|8B RL|
> |-|-|-|-|-|-|
> |Undefined function calls|0|1|0|1|0|
> |Duplicate function definitions|0|2|1|0|0|
> |Unused imports|0|2|1|0|0|
> |Initialization|Concentric rings|Random|Grid|Hexagonal multi-start|Hexagonal multi-start|
> |Counterproductive mutations|0|3 (safety factor, shrinking, overwrites)|0|1|0|
> |Optimization|Greedy radii only|Greedy radii (counterproductive safety factor)|Greedy radii (well-tuned margins)|scipy SLSQP (buggy constraint)|scipy SLSQP (clean closures, error handling)|
> **Table R2**: Strategies on Third Autocorrelation.
> |Feature|Init|1.5B Infer|1.5B RL|8B Infer|8B RL|
> |-|-|-|-|-|-|
> |Construction strategy|Gaussian + 2 freq. components|Complex wavelet (broken)|Multi-frequency|Kaiser window + cosine bases|Windowed cosine|
> |Optimization method|NA|NA|NA|SLSQP (single method)|3 methods (L-BFGS-B, TNC, SLSQP), keeps best|
> |Escape heuristic|NA|NA|NA|NA|Simulated annealing (sign-flip perturbation with temperature decay)|
>
>
> **Table R3**: Strategies on Second Autocorrelation.
> |Feature|Init|8B Infer|8B RL|
> |-|-|-|-|
> |$L_{\infty}$ gradient|Subgradient|Smooth max|Smooth max (proper log-sum-exp)|
> |Gradient preconditioning|NA|NA|Curvature-aware via FFT|
> |Structured init|Random + elitist respawn|Comb signal + standard candidates|Ansatz candidates (broad envelope + asymmetric spikes)|
> |Exploration mechanism|Gradient noise|Comb perturbation|Asymmetric gradient damping|
> |Broken code paths|0|2 (overwrite, wrong FFT indexing)|0|
>
>
> > **Q3**: lacks a rigorous analysis of diversity evolves during training… any mode collapse or reduced exploration under RL?
>
>
> **A3**: Thanks for this helpful suggestion. To study this, we use island code diversity metric from OpenEvolve, which computes a fast pairwise diversity score based on differences in length, line count, and character set. We ran two additional experiments on CirclePacking with ProRL-1.5B as in Table R4.
>
> **Table R4**
> |Step|RL score|Infer score|RL Diversity|Infer Diversity|
> |-|-|-|-|-|
> |0|0.96|0.96|535|535|
> |10|1.55|1.44|927|555|
> |40|1.96|1.88|114|61|
> |100|2.26|2.03|158|83|
> |150|2.32|2.03|38|104|
>
> Before step 100, the RL run does not show lower diversity than inference. After that, RL diversity decreases substantially, while its score remains much higher than inference. This suggests that RL does not simply collapse early to a narrow region.
> We also inspected the evolving trajectories qualitatively. RL covers most of the major paradigm shifts that also appear in the inference run, but tends to combine and exploit useful strategies more effectively. By contrast, inference maintains higher late-stage diversity partly through many random and useless mutations.
> Nevertheless, we agree that it is a general challenge of few exploration when applying RL to difficult open problems with very sparse positive signals, and we view it as an important future direction.

---

> > ### Author Rebuttal · Reviewer_L9JM · 2026-04-03
> >
> > Thanks for the additional analysis.
> >
> > **Remaining concern:** my question regarding Fig. 3 is about why inference-only remains competitive in the early stage, which seems practically important under limited compute. From a practical perspective, are there task characteristics or early signals that predict when RL will provide a meaningful advantage over inference-only

---

> > > ### Author Response · Authors · 2026-04-04
> > >
> > > Thanks for your recognition and the follow-up question. We first want to clarify the interpretation of Fig. 3.
> > >
> > > There are three curves in Fig. 3:
> > >
> > > (1) **Blue**: ThetaEvolve with inference-only on the original base model.
> > >
> > > (2) **Orange**: ThetaEvolve with RL training starting from the same base model.
> > >
> > > (3) **Green**: inference-only again, but using the step-150 checkpoint from the best RL run on CirclePacking (best score 2.5225, as noted in Line 335), rather than the original base model.
> > >
> > > Therefore, if your concern is that the green curve (3) is competitive with or better than the orange curve (2) in the early stage, **we believe this is actually evidence that RL has already taught the model useful behaviors, such as format correction, fewer breaking changes, and stronger search strategies**. Since the green curve uses an RL-trained checkpoint, it can start with these learned capabilities already internalized, while the orange curve is still training from the original base model and must acquire them during the run. This is also why the green curve can improve quickly even on tasks different from the one used for RL training. More details are discussed in Sec. 4.3.1.
> > >
> > > If instead you mean the comparison between the blue (1) and orange (2) curves, we agree that inference-only can sometimes get similar performance at the very beginning. However, this usually only happens in the earliest stage  (e.g., first 20 steps in 600 steps). After that, RL tends to perform better, especially in the average score (Mean-over-seeds). For a single best run, sometimes inference may occasionally look better at longer early stages (e.g., ThirdAutoCorrIneq, Best seed), which we believe **is partly due to run-to-run randomness** and may happen more often on optimization-heavy tasks as AutoCorrelation Inequality, e.g., a good searching strategy is much more important than initialization, and thus in a specific run that model try to improve optimization first rather than initialization would get advantage more quickly. Overall, based on the average results, we do not think RL is worse than inference-only in the early stage when both start from the same base model. Rather, RL is initially similar and then becomes better once it learns behaviors such as correct formatting, non-breaking edits, and stronger strategy composition.
> > >
> > > Thanks again for your question and we would revise our paper accordingly to make it more clear.

---

### Official Review · Reviewer_vZPa · 2026-04-04

**Soundness:** 3
**Presentation:** 3
**Significance:** 3
**Originality:** 3
**Overall Recommendation:** 4
**Confidence:** 2

**Summary:**

This paper introduces ThetaEvolve, a novel Test-time Reinforcement Learning framework tailored for reasoning problems. By reconstructing the RL environment based on single-LLM batch sampling and a large-scale dynamic program database, this work substantially lowers the computational threshold by eliminating the reliance on expensive ensembles of frontier models.

**Compliance With Llm Reviewing Policy:**

Affirmed.

**Final Justification:**

Based on the current version of the manuscript, I recommend a "weak accept". If the authors adequately address my concerns in the revision, I would be open to raising my score.

**Key Questions For Authors:**

See weaknesses.

**Limitations:**

Please discuss the generalization ability of ThetaEvolve.

**Strengths And Weaknesses:**

## Strengths

1. **Dynamic verifiable RL environment for capability internalization and enhancement:** The paper effectively integrates reinforcement learning into the code evolution process by constructing a dynamic and verifiable environment. This allows the model to continuously sample high-quality and efficient parent programs from the database for further optimization. Such a mechanism not only expands the search space effectively, but also helps the model internalize evolutionary strategies, thereby improving its ability to tackle complex open problems.
2. **Single-LLM batch processing architecture with strong efficiency advantages:** By leveraging an 8B open-source model, the framework substantially improves the throughput of both inference and evolution while lowering the computational barrier for research teams exploring frontier mathematical problems. This makes the approach relatively accessible and community-friendly.

## Weaknesses

- **Conceptual novelty is moderate:** Many components appear to be careful engineering adaptations of AlphaEvolve, OpenEvolve, and standard RL machinery. As a result, the main contribution seems to lie more in integration and scaling than in a sharply novel algorithmic idea.
- **Evaluation scope is narrow and somewhat uneven:** The paper evaluates the method on only a small set of open mathematical optimization tasks with handcrafted verifiers, and not all tasks receive the full RL treatment. This limits confidence in the broader generality of the approach.
- **Heavy reliance on task-specific scaffolding:** The method requires manually designed evaluators, initial programs, meta-information, and tuned reward-shaping parameters. In addition, for several tasks, prompt insights are generated using GPT-5 and Claude 4.5, which weakens the claim of a fully self-contained open-source pipeline.
- **Some headline comparisons are not fully apples-to-apples:** The strongest CirclePacking result is reported under the OpenEvolve tolerance-based setting. Although the paper argues that the improvement transfers to the stricter setting, this still makes direct comparison with AlphaEvolve’s reported bound less clean.
- **The value of RL over additional inference compute is not always decisive:** In some cases, especially with Distill-Qwen3-8B, longer inference-only runs come close to the RL results. The paper therefore does not fully quantify when the additional complexity and compute cost of RL are truly justified.

---

### Decision · Program_Chairs · 2026-04-30

**Decision:**

Accept (regular)

**Comment:**

The paper presents an extension of AlphaEvolve that uses small LLMs for mathematical discovery by allowing test-time learning. Reviewers note the small model efficiency, strong empirical results, cross-task generalizability and good engineering but also note sample inefficiency and moderate novelty. All in all the paper is a solid contribution.